# Invariance-based Learning of Latent Dynamics

**Kai Lagemann**[*]
Statistics and Machine Learning, German Center for Neurodegenerative Diseases (DZNE), Bonn, Germany
`kai.lagemann@dzne.de`

**Christian Lagemann**[*]
Department of Mechanical Engineering, University of Washington, Seattle, USA

**Sach Mukherjee**
Statistics and Machine Learning, German Center for Neurodegenerative Diseases (DZNE), Bonn, Germany
MRC Biostatistics Unit, University of Cambridge, Cambridge, UK
`sach.mukherjee@dzne.de`

## Abstract

We propose a new model class aimed at predicting dynamical trajectories from high-dimensional empirical data. This is done by combining variational autoencoders and (spatio-)temporal transformers within a framework designed to enforce certain scientifically-motivated invariances. The models allow inference of system behavior at any continuous time and generalization well beyond the data distributions seen during training. Furthermore, the models do not require an explicit neural ODE formulation, making them efficient and highly scalable in practice. We study behavior through simple theoretical analyses and extensive empirical experiments. The latter investigate the ability to predict the trajectories of complicated systems based on finite data and show that the proposed approaches can outperform existing neural-dynamical models. We study also more general inductive bias in the context of transfer to data obtained under entirely novel system interventions. Overall, our results provide a new framework for efficiently learning complicated dynamics in a data-driven manner, with potential applications in a wide range of fields including physics, biology, and engineering.

## 1 Introduction

Dynamical models are central to our ability to understand and predict natural and engineered systems. A key question in studying dynamical systems is predicting future behavior. Real-world systems often show time-varying behavior that is much too complex for straightforward statistical forecasting or extrapolation approaches. This is due to the fact that the temporal behavior, while potentially explained by an underlying dynamical model, can show strong, possibly abrupt changes in the observation/data space, precluding effective modeling via traditional curve-fitting or extrapolation. Furthermore, different instances or realizations of a single scientific/engineering system (e.g. with different initial conditions or constants) can show large differences in terms of data distributions, hence going beyond standard in-distribution assumptions of traditional data-fitting approaches.

Against this background, in recent years a wide range of sophisticated dynamical machine learning approaches have been proposed, including in particular neural ordinary differential equations Chen et al. (2018) and a wider class of related models (see for example Zhi et al. (2022); Finlay et al. (2020); Duong and Atanasov (2021); Choi et al. (2022); Chen et al. (2021); Kim et al. (2021b)). Broadly speaking, these models go beyond simple curve-fitting/extrapolation schemes by leveraging suitable inductive biases to allow learning of latent dynamical models. There has been rapid progress in this area but key challenges remain for complicated real-world systems, due to multiple factors, including data limitations, generalization to unseen settings, irregular time sampling and issues relating to long-horizon trajectories (Iakovlev et al., 2023).

---

[*]These authors contributed equally to this work.

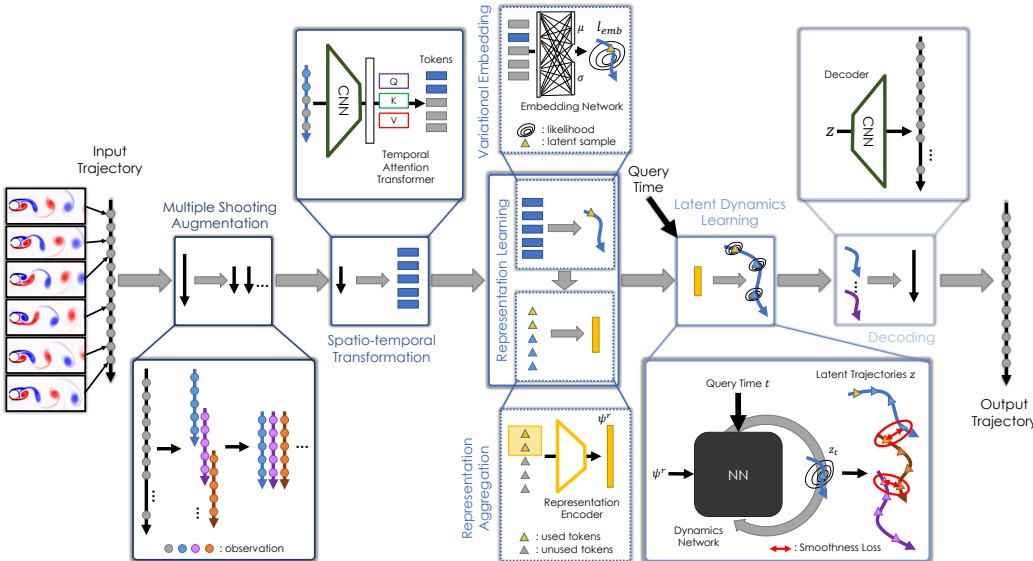

Figure 1: Architecture: Learning of Latent Dynamics via Invariant Decomposition (LaDID). A set of high-dimensional snapshots on a regular or irregular time-grid serves as the input to LaDID. The trajectory is split into subpatches using Multiple Shooting Augmentation (i). The first time-points of each subpatch are used to compute a subtrajectory representation: features of the selected snapshots are re-weighted w.r.t. time and spatial location (ii), transformed to a low-dimensional embedding (iii), and aggregated into one trajectory representation $\psi^r$ (iv). During inference, the latent dynamical model is conditioned on the representation $\psi^r$. Prediction is possible at any continuous time by querying the latent state of any time point (v). Latent subtrajectories are sewn together by a smoothness loss. Finally, the entire latent trajectory is decoded to the observation space (vi).

Motivated by these challenges, we propose a new framework, called "Latent Dynamics via Invariant Decomposition" or LaDID, for learning latent dynamics from empirical data. LaDID leverages certain scientifically-motivated invariances to permit efficient learning and effective generalization. In numerous real-world dynamical systems, longitudinal trajectories may exhibit significant variability, e.g. due to differences in initial conditions or model constants. Each temporal trajectory, which we refer to as a "realization", represents a particular manifestation of the system's dynamics under certain conditions. A key notion underpinning LaDID is that, even when temporal trajectories from a class of scientific systems appear diverse, they can still be effectively explained by an appropriate, in a sense "universal", model; such a model is therefore *realization-invariant*. To facilitate broad generalization, LaDID introduces factors specific to each realization as inputs to its universal model. These factors are hence *realization-specific* and play the role of (implicitly) encoding aspects such as the initial states of the system or specific model constants. A transformer-based architecture is used to learn all model aspects from data, including both *realization-specific* (RS) and *realization-invariant* (RI) information. At inference-time LaDID can output predictions for *any* continuous time. Due to the universal nature of the RI model, LaDID can effectively handle substantial variations in system behavior and data distributions (e.g. due to changes in initial conditions or system constants). We empirically validate LaDID on various spatio-temporal systems with dynamics on regular and irregular time grids governed by ordinary or partial differential equations. The LaDID architecture is fast and easy to train and, as we show, substantially outperforms existing neural-dynamical models over a range of challenging tasks. Thus, our main contributions are:

- We present a novel framework, and associated transformer-based network, for the separation of realization-specific information and (realization-invariant) latent dynamical systems.

- We systematically study performance on short- and longer-horizon prediction of a wide range of complex temporal and spatio-temporal problems, comparing against a range of state-of-the-art neural-dynamical baselines.

- We study the challenging case of transfer to data obtained under entirely novel system interventions via a few-shot learning (FSL) approach.

## 2 RELATED WORK

Flexible neural models have been exploited to learn dynamical models, with connections drawn between deep architectures and numerical solvers for ODEs, PDEs and SDEs (Chen et al., 2018; Weinan, 2017; Lu et al., 2018; Ruthotto and Haber, 2020; Haber and Ruthotto, 2017; Richter-Powell et al., 2022). Algorithms rooted in neural differential equations (NODEs) have been shown to offer benefits relative to standard recurrent neural networks (RNNs) and their variants. However, since NODEs directly relate to the problem formulation of standard ODEs, they inherit some associated limitations. Specifically, the temporal dynamics only depend on the current state but not on the history putting a limit on the complexity of the trajectories that NODEs can model (Holt et al., 2022).

Improvements have been proposed that augment the latent state space to broaden the range of dynamical systems that can be learned (Dupont et al., 2019) while Rubanova et al. (2019) suggested a combination with an autoregressive RNN updated at irregularly sampled time points. Complementary work has proposed neural controlled differential equations, a mechanism to adjust the trajectory based on subsequent observation (Kidger et al., 2020; Morrill et al., 2021). Massaroli et al. (2021) transferred the concept of multiple shooting to solve differential equations to the conceptual space of NODEs and Iakovlev et al. (2023) extended this concept to sparse Bayesian multiple shooting, with both works evaluating latent NODEs. However, for certain types of dynamics numerical instability poses challenges for NODEs (and their variants) (Li et al., 2020). This is due to the fact that NODEs rely on numerical ODE solvers to predict the latent trajectory (forward pass) and this becomes unstable with longer time horizons (Iakovlev et al., 2023). In contrast, by exploiting RS and RI invariances our model eschews explicit neural ODEs altogether, providing an arguably simpler and faster transformer-based scheme that can be trained in a straightforward fashion, as we detail below.

The idea of leveraging invariances is a core notion in scientific modeling and is seen throughout the natural sciences at a conceptual and practical level. For instance, in the field of AI, it has been utilized in the context of video frame prediction as demonstrated by various studies (van der Wilk et al., 2018; Franceschi et al., 2020; Kabra et al., 2021). LaDID differs from these approaches because it uses invariances to model a kind of generalized initial condition (motivated by scientific uses of dynamical models; see below) and because it learns continuous latent trajectories (as opposed to an autoregressive model), including in the irregular time-sampling case. See Section A of the Appendix for further details on related work.

## 3 PROBLEM STATEMENT AND CONCEPTUAL OUTLINE

We start with a high-level problem statement and outline the motivating concepts behind LaDID, deferring a detailed description of the architecture itself to subsequent Sections. We focus on settings in which we have (finite) observations of a system of interest at time points $t \in T$ (potentially irregularly spaced). We do not require any specific prior information on the underlying model; rather our approach is data-driven, informed by certain very general invariances as detailed below.

For an instance/realization $r$ of a dynamical system of interest let $X_r \in \mathbb{R}^{T \times C \times H \times W}$ denote a high-dimensional trajectory in the observation space; $T$ denotes the number of time steps (possibly irregular), and $C$, $H$, and $W$ respectively the number of channels, frame height and frame width (in empirical examples we focus on image-like data inputs) of the observations. Let $X = (X_r)_{r \in R}$ denote the collection of available training trajectories; the notation emphasizes the possibility that available data spans multiple instances/realizations. Given these initial data, LaDID seeks to predict future observations $x_t^r$ for any continuous time $t$ and for any realization $r$.

**From invariances to a simple learning framework.** We start by studying a basic, conceptual set-up that sheds light on how our assumptions lead to a simple, but very general, learning framework. Intuitively, we demonstrate that while we cannot guarantee recovery of the true underlying model parameters, under mild invariance assumptions, there exists a function capable of reconstructing the true observations, even when dealing with potentially highly heterogeneous parameters and data. Importantly, we do not make any prior assumptions about the nature of these potentially complex and nonlinear functions. Instead, our learning framework simultaneously uncovers and refines these functions in a data-driven, end-to-end manner, as elaborated in Section 4.

Consider an entirely general system $f$ in which some aspects are realization-specific (RS) while others are realization-invariant (RI); the latter model aspects are general to *all* instances/realizations of the model, while the former are not. We assume also that the RS aspects are the same for all

times; i.e. these are time-invariant. Our model is not specific to physical ODE-like models, but rather generalizes invariances to permit flexible learning in a wide range of settings. To fix ideas, it is nevertheless useful to consider the specific example of a physical model class described by ODEs. For this latter setting, the model itself would be RI, while the initial conditions or system constants could be thought of as RS. As a result, the same model class can describe a wide range of actual systems which share an underlying scientific basis while differing (perhaps strongly) in details.

Let $x_t^r = f(t; \Theta_r)$ denote the fully general model. Here, $\Theta_r$ is the complete parameter set needed to specify the time-evolution;, including both RS and RI parts. To make the separation clear, we write the two parts separately as $x_t^r = f(t; \theta_r, \theta)$, where $\theta_r$, $\theta$ are respectively the RS and RI parameters and $r$ indexes the realization. Suppose $\hat{\theta}_r$ is a candidate encoding of the RS information. We now assume that the encoding, while possibly incorrect (i.e. such that $\hat{\theta}_r \neq \theta_r$) satisfies the property $\exists m, \exists \theta_m : \theta_r = m(\hat{\theta}_r; \theta, \theta_m)$, where $m$ is a function that "corrects" $\hat{\theta}_r$ to give the correct RS parameter. This essentially demands that while the encoding $\hat{\theta}_r$ might be very different from the true value (and may even diverge from it in a complicated way that depends on unknown system parameters), there exists an RI transformation that recovers the true RS parameter from it, and in this sense the encoding contains all RS information. We call this the *sufficient encoding assumption* (SEA). In the context of dynamics involving latent variables $z \in \mathbb{R}^q$ again consider a model with RS and RI parts but at the level of the latents, i.e. $z_t^r = f(t; \theta_r, \theta)$. Assume the observables are given by (an unknown) function $g$ of the hidden state $z$. Then, as shown in the Appendix B, assumptions similar in spirit to SEA above imply existence of a mapping function that allows arbitrary queries to in principle be predicted via a straightforward learning scheme. In brief, these assumptions[*] require that we have access to an approximation $\hat{g}$ of the observation process that while possibly very far away from the true function, can nonetheless be "corrected" in a specific way (see Appendix B for details). We emphasize that as for SEA, it is *not* required that we have a good approximation in the usual sense of being close to the true function $g$, only that there exists a correction of a certain form. Importantly, at no point do we actually require prior knowledge of the underlying dynamical system, its true latent variables nor any system constants or initial conditions; rather, the assumptions imply existence of mapping functions that can be learned from data, even when the underlying model is entirely unknown at the outset.

**Analogy to traditional ODE formulations.** To facilitate a more intuitive understanding of our framework, we discuss further analogies to standard ODE problems. Typical ODE solvers comprise a formulated ODE function and some initial values (IVs) which are evolved over time using well-known integration methods, e.g, Euler, Runge-Kutta, or DOPRI schemes. From a high-level perspective, the IVs relate to our RS representation while the ODE functions and its corresponding integration approach is our RI part. However, please note that this analogy only holds on a superficial level due to two fundamental differences: First, LaDID relies only on *observations* of a specific system, for which underlying state variables required to apply standard ODE solvers are not known (and in fact not observed). Hence, our RS representation benefits from access to a collection of high-dimensional observations. Second, our RI function is continuous in time and therefore unites temporal integration and dynamics function on an abstract level. To do so, we condition our RI model on specific RS representations and only query time points from it to obtain a discretized latent trajectory.

## 4 METHODS

Based on these initial arguments, we now put forward a specific architecture to allow learning in practice. At a high level, the architecture implements the general mapping approach outlined above (further details in the Appendix), learning, in a data-driven manner, both RS and RI model components and putting these together to allow prediction at any continuous time in a query realization. Implementation details and a full ELBO derivation can be found in Section C - F of the Appendix.

### 4.1 MODEL, INFERENCE AND PREDICTION

**Model.** The LaDID architecture[*] is composed of three main components: the encoder $f_{\phi_{enc}}$, the invariant dynamical system $f_{\phi_{dyn}}$, and the decoder $f_{\phi_{dec}}$, respectively governed by parameters $\phi_{enc}$, $\phi_{dyn}$, and $\phi_{dec}$. The encoder is a collection of three NNs: a CNN processing spatial information in

---

[*]Called the *sufficient mapping assumption* (SMA) and realization-specific SMA (RS-SMA) for the cases in which the latent-to-observed mapping is itself RI or RS respectively; see Appendix for details.

[*]A reference implementation is available under `https://github.com/kl844477/LaDID`.

the observation space, a transformer utilizing temporal attention and a learnable mapping function. Since we want to predict future observations based on a few observations, we only use the first $K$ datapoints in time and process these in a shared convolutional encoder (green trapezoid in Figure 1 (ii)). We employ a shallow CNN that compresses the input to 1/16 of the initial input size using four ReLU activated and batch-normalized convolutional layers. The resulting tensors are then flattened and mapped linearly to a single vector. Next, we use a transformer on the $K$ output vectors of the convolutional encoder, applying temporal attention to reweigh vectors. We tested two approaches (Bulat et al., 2021; Iakovlev et al., 2023) with comparable performance which are discussed in the Appendix in more detail. For each of the $k \in K$ time aware representations $\rho_k^{TA}$, we sample a latent embedding using the reparameterization trick, i.e. $l_k^{emb} \sim \mathcal{N}(f_\mu(\rho_k^{TA}), f_\sigma(\rho_k^{TA}))$. The final trajectory representation $\psi^r$ is the output of an aggregation over all $K$ tokens. In our implementation, we choose a simple yet effective mean-aggregation which can be changed based on the task at hand. The second important part of our proposed framework is the dynamical model $f_{\phi_{dyn}}$. We utilized a three layer MLP which can also be interchanged by other functions. To obtain a latent trajectory, we condition the latent dynamical model on our end-to-end learned trajectory representation $\psi^r$ and roll-out the latent trajectory $z$ based on the queried time points $t_q$ represented through a time encoding which we choose as a set of different sine and cosine waves with different wave length. Finally, we map all data points of our latent trajectory back to the original observation space. Our decoder module $f_{\phi_{dec}}$ is kept very simple consisting of four deconvolutional layers.

The key novelty of our approach lies in the unique structure of the latent space mimicking the interplay of realization-specific information and a realization-invariant dynamical model similar to the frame of differential equations. However, we can significantly reduce computational costs as we are not forced to solve explicitly any differential equation since we rely on an effective end-to-end learning scheme.

**Generative model, inference and optimization.**  We now turn the descriptive technical context of our method to a probabilistic model. Our graphical model (see Figure A1 in the Appendix) consists of trainable parameters $\Phi = \phi_{enc} \cup \phi_{dyn} \cup \phi_{dec}$, a random variable $\psi^r$ which additionally acts as global random variable at the level of latent states $z_{t_q}$ and observations $x_{t_q}$. The index $t_q$ refers to a specific queried time point within a trajectory. The joint distribution is given by

$$p(x, z, \psi^r) = p(x, z|\psi^r)p(\psi^r) = p(x|z)p(z|\psi^r)p(\psi^r). \tag{1}$$

Our graphical model assumes these independencies: (i) The dataset contains i.i.d. trajectories of varying length. (ii) The observation of trajectory $x_{t_q}^r$ at time $t_q$ is conditionally independent of $x_{t_{q-1}}^r$ at time $t_{q-1}$, given latent states $z_{t_q}^r$ and trajectory representation $\psi^r$: $p(x_{t_q}|z_{t_q}, \psi^r) \perp\!\!\!\perp p(x_{t_{q-1}}|z_{t_{q-1}}, \psi^r)$. Analyzing data with this graphical model involves computing posterior distributions of hidden variables given observations

$$p(z, \psi^r|x) = \frac{p(x, z, \psi^r)}{\int p(x|z)p(z|\psi^r)p(\psi^r)dzd\psi^r}. \tag{2}$$

To effectively process long-horizon time series data, we apply a variant of *multiple shooting*. However, since our model does not rely on an explicit ODE formulation, we are not concerned with turning an initial value problem into a boundary value problem (Massaroli et al., 2021). Instead, we incorporate a Bayesian continuity prior (Hegde et al., 2022; Iakovlev et al., 2023) to extend the multiple-shooting framework from deterministic neural ODEs to a probabilistic context. Our approach dissects each realization $x_{t:T}^r$ into a series of $N$ overlapping subtrajectories and independently condenses each patch into a latent representation. Within this Bayesian multiple shooting framework, the smoothness prior connects the patches via

$$p(z|\psi^r) = \prod_{i=1}^{N} p(z_n|\psi_n^r)p(z_n|z_{n-1}, \psi_{n-1}^r) \tag{3}$$

to form a cohesive global trajectory. We leverage the independence of trajectory representations in subpatches i.e. $p(z_i|\psi_i^r) \perp\!\!\!\perp p(z_j|\psi_j^r)$. For the continuity prior, we follow Hegde et al. (2022) and place a Gaussian prior on the error between consecutive subtrajectories, i.e. $\Delta \sim \mathcal{N}(0, \sigma_\Delta)$ entailing exact overlapping if $\Delta \to 0$. This yields our continuity prior

$$p(z_n|z_{n-1}, \psi_{n-1}^r) = \mathcal{N}((z_n^{t_1}|z_{n-1}^{-t}, \psi_{n-1}^r), \sigma_\Delta), \tag{4}$$

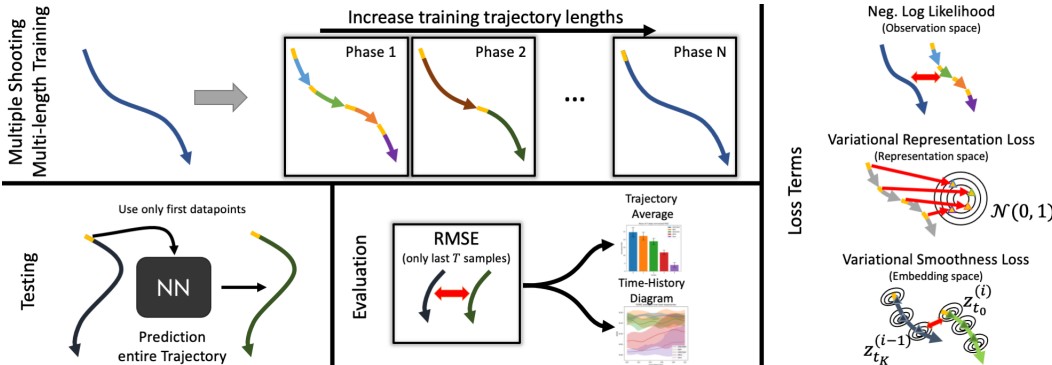

Figure 2: Training scheme with losses, and test/evaluation procedure. Top left: Multiple Shooting Multi-length Training. An input trajectory is split into subpatches. Subtrajectory length is increased in multiple phases. Bottom left: Testing: only the first few points are used to roll-out the latent trajectory and transformed to the observational space. Evaluation: Last $T$ samples of the predicted trajectory are used to compute the evaluation metrics, the average of the summed normalized mean squared error and a time history diagram showing the error evolution. Right: Loss consisting of three parts: negative log likelihood loss to penalize reconstruction errors , representation loss to define a gradient field between representations, smoothness loss to penalize jumps between latent subpatches.

where the time index $-t$ refers to the last time point of a subpatch. The prior trajectory representation is set to a Gaussian, i.e. $p(\psi^r) \sim \mathcal{N}(0,1)$. With the priors introduced above, we get the following generative model (we drop the subpatch index $n$ for improved readability):

$$p(l_K^{emb}|x) = \mathcal{N}(f_{\phi_{enc},\mu}(x_K), f_{\phi_{enc},\sigma}(x_K)) \tag{5}$$

$$p(\psi^r|x) = f_{agg}(l_K^{emb}) \tag{6}$$

$$p(z|\psi^r, x) = f_{\phi_{dyn}}(\psi^r, t_q) \tag{7}$$

$$p(x|z) = \mathcal{N}(f_{\phi_{dec}}(z), \sigma_{dec}) \tag{8}$$

For inference, we use Gaussian approximations and set $\sigma_{dec} = 10^{-2}$. We then seek to minimize the KL divergence $\text{KL}[q(z, \psi^r)||p(z, \psi^r|x)]$, essentially equivalent to maximizing the ELBO in eq. 9. A full derivation of this ELBO can be found in Section C of the Appendix.

$$\max \mathbb{E}_{q(z,\psi^r)} \underbrace{\sum_{n=1}^{N} \ln p_n(\hat{x}_n)}_{\text{(i) likelihood}} - \underbrace{\sum_{n=1}^{N} \text{KL}(q(\psi_n^r)||p(\psi_n^r|x_n))}_{\text{(ii) representation prior}} - \underbrace{\sum_{n=2}^{N} \mathbb{E}_{q(z,\psi^r)} \text{KL}(q(z_n)||p(z_n|z_{n-1}, \psi_n^r))}_{\text{(iii) smoothness prior}} \tag{9}$$

## 5 EXPERIMENTAL SET-UP

Experiments are structured into four different series that shed light on the performance of LaDID. We provide a short overview of the experimental set-up in the following. Further details can be found in Section J in the Appendix.

**Datasets.** We consider a wide range of physical systems ranging from relatively simple ODE-based datasets to complex turbulence-driven fluid flows. Specifically, we consider high-dimensional observations ($p$=16, 384) from: a nonlinear swinging pendulum; a swinging double pendulum; realistic simulations of the two-dimensional wave equation; a lambda-omega reaction-diffusion system; the two-dimensional incompressible Navier-Stokes equations; and the fluid flow around a blunt body solved via the latticed Boltzmann equations. This extensive range sheds light on performance on complex datasets relevant to real-world use-cases, including models frequently used in the literature on dynamical modeling. Regular and irregular time grids are included. We study also the challenging problem of making predictions in a completely novel setting obtained by *intervention* on the system. This is similar in spirit to experiments seen in causal AI and in this case involves generating small datasets on *intervened* dynamical systems (either via modifying the underlying systems, for example by changing the gravitational constant or the mass of a pendulum, or via augmenting the realization-specific observation, e.g. by changing the length of a pendulum or the

location of a simulated cylinder) and fine-tuning a pre-trained model on a fraction of the data in the target setting. We direct the interested reader to Appendix I for further details.

**Training.** Training is carried out in a multi-phase schedule w.r.t. the multiple shooting loss in eq. 9. In the different phases, we split the input trajectory into overlapping patches and start learning by predicting one step ahead. We double the number of prediction steps per patch every 3000 epochs meaning that learning is done on longer patches with decreased number of patches per trajectory (where trajectory length is not divisible by number of steps, we omit the last patch and scale the loss accordingly). In the final phase, training is carried out on the entire trajectory. All network architectures are implemented in the open source framework PyTorch (Paszke et al., 2019). Further training details and hyperparameters can be found in Appendix F.

**Testing.** We test the trained models on entirely unseen trajectories. During testing, the first $k=10$ trajectory points are provided to the trained model. Based on these samples, an RS representation $\psi^r$ is computed and used to roll out the trajectory to the time points of interest. Finally, predictions and ground truth observations are compared using the evaluation metrics below.

**Evaluation metrics.** We consider mean squared error (MSE) over trajectories: inference runs over $2T$ steps with MSE computed over the *last* $T$ timesteps, allowing assessment for relatively distant times (relative to the reconstruction MSE). We set $T = 60$ for all experiments, with MSE normalized w.r.t. average (true) intensity, as recommended in Zhong et al. (2021); Botev et al. (2021). Additionally, we provide time history diagrams plotting root mean square error (RMSE) against normalized time (mapping the interval $[T, 2T]$ to the unit interval). Metrics are averaged across all test trajectories and five runs, with mean and $75\%$ inter-quantile ranges (IQR) reported. Subsampled predictions and pixelwise $L_2$ error of one (randomly chosen) trajectory is shown for visual inspection; we acknowledge that these cannot always be representative and should be considered alongside formal metrics. See Figure 2 for intuition on the train/test procedure and metrics.

**Comparisons.** We compare our approach to recent models from the literature, including ODE-RNN (Rubanova et al., 2019), NDP (Norcliffe et al., 2021), ODE2VAE (Yildiz et al., 2019), and MSVI (Iakovlev et al., 2023). In common with LaDID, these models feature encode-simulate-decode structures and seek to learn low-dimensional latent dynamics. ODE2VAE simulates latent trajectories in a straightforward fashion using a BNN to model the underlying dynamics. In contrast, ODE-RNN, NDP and MSVI leverage neural ODE solvers to integrate latent states forward in time. Further details regarding these baselines can be found in Section G of the Appendix.

# 6 RESULTS

First, we examined performance on synthetic data for which the training and test data come from the same dynamical system. This body of experiments test whether the model can learn to map from a finite, empirical dataset to an effective latent dynamical model. Second, we examine few-shot generalization to data obtained from systems subject to nontrivial intervention (and in that sense strongly out-of-distribution). In particular, we train our model on a set of trajectories under interventions, i.e. interventions upon the mass or length of the pendulum, changes to the Reynolds number, or variations to the camera view on the observed system, and apply the learned inductive bias to new and unseen interventional regimes in a few-shot learning setting. This tests the hypothesis that the inductive bias of our learned latent dynamical models can be a useful proxy for dynamical systems exposed to a number of interventions.

## 6.1 BENCHMARK COMPARISONS TO STATE-OF-THE-ART MODELS FOR ODE AND PDE PROBLEMS

We begin by investigating whether LaDID can learn latent dynamical models in the conventional case in which the training and test data come from the same system. We evaluate the performance of ODE-RNN, ODE2VAE, NODEP, MSVI and LaDID on the data described in Section 5 and Section I of the Appendix with increasing order of difficulty, starting with the non-linear mechanical swing systems with underlying ODEs, before moving to non-linear cases based on PDEs (reaction-diffusion system, 2D wave equation, von Kármán vortex street at the transition from laminar to turbulent flows, and Naiver-Stokes equations). Due to limited space, we only present results for a subset of performed experiments but refer the interested reader to Appendix K for a detailed presentation of all results.

**Applications to ODE-based systems.** For visual inspection and intuition, Figure 4 provides predicted observations $\hat{x}_t^\tau$ of a few time points of one test trajectory of the single pendulum dataset for all tested algorithms, followed by the ground truth trajectory and the pixelwise $L_2$-error. In addition, Figure 3

presents the normalized MSE over entire trajectories averaged across the entire test dataset and the evolution of the RMSE over time for the second half of the predicted observations averaged over all test trajectories (see Section 5) is provided in the Appendix. Across all ODE-based datasets LaDID achieves the lowest normalized MSE. The time history diagram (see Figure K.1 in the Appendix) reveals gains using LaDID for long-horizon predictions relative to all other algorithms tested.

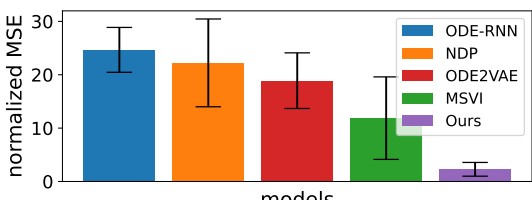

This can also seen by visual inspection in Figure 4 as for other approaches the predicted states at later time points deviate from the ground truth trajectory substantially while LaDID's predictions essentially follow the ground truth. Considering only the baselines, one can observe that MSVI (a sophisticated, recently proposed approach), predicts accurately within a short-term horizon but nonetheless fails on long-horizon predictions. The results for the challenging double pendulum test case can be found in the Appendix.

Figure 3: Test errors - Normalized MSE

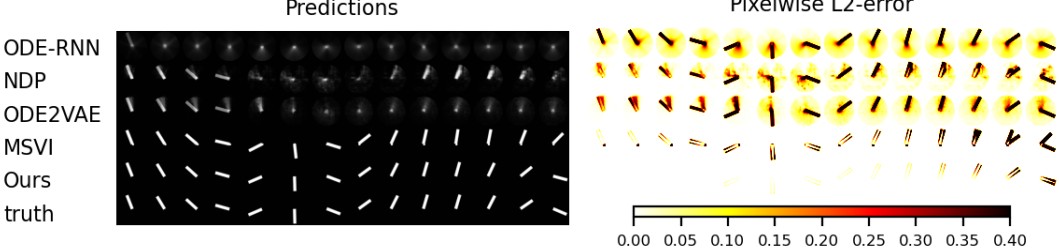

Figure 4: Left: predicted test trajectory at various timesteps $t$, right: corresponding pixelwise error.

**Applications to PDE-based processes.** We additionally evaluated all baselines and our proposed method on PDE-based processes. Due to space restrictions, we focus our analysis on the flow evolution characterized by the Navier-Stokes equation in the two dimensional case, which is of great importance in many engineering tasks, e.g. the analysis of internal air flow in a combustion engine (Lagemann et al., 2022), drag reduction concepts in the transportation and energy sector (Gowree et al., 2018; Lagemann et al., 2023a; Mäteling et al., 2023), and many more. Results in Figure 5 show that LaDID clearly outperforms all considered comparators. The normalized MSE is the lowest and the averaged RMSE is also the lowest at any time. This is echoed in the other experiments whose results are presented in detail in Section K in the Appendix.

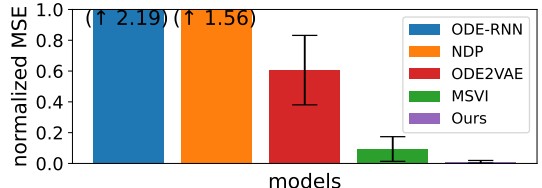

Figure 5: Test error - Normalized MSE

Overall, these results support the notion that LaDID achieves good performance for challenging ODE and PDE based systems. We direct the interested reader to Section K for the complete collection of experimental results supporting this statement. In this context, Table H.2 of the Appendix highlights the massively reduced computational ressources required during training and inference since LaDID eschews an explicit neural ODE formulation (including the costly ODE solvers), making it efficient and highly scalable in practice.

**Performance on regular and irregular time grids.** Here, we study the performance of LaDID on regular and irregular time grids and compare it to other neural-dynamical models (which are able to deal with irregular time series data). As shown in Figure L.1 and Figure L.2 in the Appendix, the proposed LaDID performs very similarly on both types of the time grids relative to both ODE-based benchmark examples and challenging PDE-based real-world systems, outperforming existing methods demonstrating strong and robust performance on irregularly sampled data.

**Effects of relevant network modules.** LaDID leverages three key features: a reconstruction embedding, a spatio-temporal attention module and a specifically designed loss heuristic to learn temporal dynamics from empirical data. We investigated the importance of these modules (results appear in the Section N of the Appendix). First, we compared LaDID with ablated counterparts, e.g.

a pure reconstruction loss and loss combinations either using the representation or smoothness loss. Overall, the proposed loss heuristic appears to stabilize training and yields the lowest MSE and IQR values. Second, we compared LaDID to counterparts trained on ablated attention modules. Empirical results underline the utility of the applied spatio-temporal attention. Finally, Table N.3 further shows the usefulness of the representation-specific encoding. This representation encoding can be thought of a learning-enhanced initial value stabilizing the temporal evolution of latent trajectory dynamics. Moreover we study the effect of restricted training trajectories on the performance of LaDID in Section K.7 of the Appendix to better understand efficiency under limited data.

### 6.2 GENERALIZING TO NOVEL SYSTEMS VIA FEW-SHOT LEARNING

Here, we assess LaDID's ability to generalize to a novel system obtained by nontrivial intervention on the system coefficients themselves (e.g., mass, length, Reynolds number). Such changes can induce large changes to data distributions and can be viewed through a causal lens (see also Appendix O). In particular, we train a dynamical model on a set of interventions and fine-tune it to new intervention regimes with only a few samples, finally evaluating performance on an entirely unseen dataset. We compare the performance of our prior-based few-shot learning model with a model trained solely on the fine-tuning dataset ("scratch-trained" model). In our first experiment, we use the single pendulum dataset and test the transferability hypothesis on fine-tuning datasets of varying sizes. The results show that the prior-based model outperforms the scratch-trained model at all fine-tuning dataset sizes, and achieves comparable performance to the model trained on the full dataset with a fine-tuning dataset size of 32%. At a fine-tuning dataset size of 8%, LaDID produces partially erroneous but still usable predictions, which are only slightly worse than the predictions of an advanced NODE based model, MSVI, trained on the full dataset. Further results including robustness to input noise and color shifts in the observation space appear in Section M in the Appendix.

Second, we investigate the effect of interventions on the observation process by testing the transferability to new observation settings on the von Kármán vortex street dataset. We re-simulate different cylinder locations (shifted the cylinder to the left, right, up, and down) and evaluate the performance under different fine-tuning dataset sizes. The results show that the prior-based model consistently outperforms the scratch-trained model and produces accurate and usable predictions under new observation conditions with a fine-tuning dataset size of as little as 8%. These findings support our hypothesis that LaDID is capable of extracting general dynamical models from training data. Additional transfer learning experiments are detailed in the Section P of the Appendix, studying the model's performance when jointly trained on a dataset encompassing Reynolds numbers of $Re = [100, 250, 500]$ and subsequently applied for zero-shot predictions on unseen $Re$ numbers.

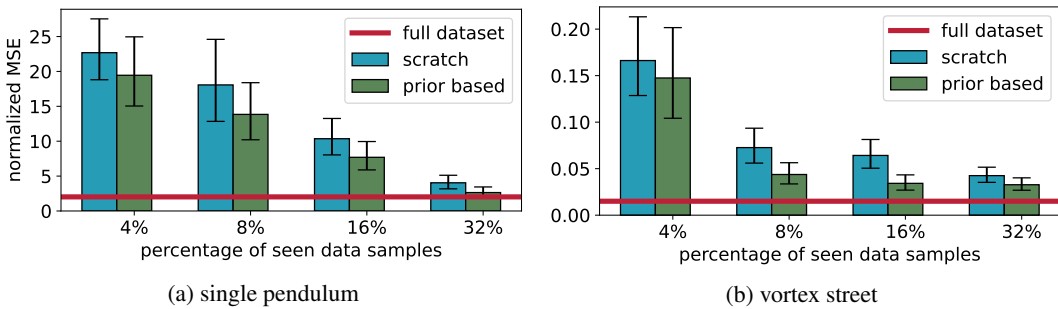

|                    |                    |
| :----------------: | :----------------: |
| (a) single pendulum | (b) vortex street |

Figure 6: Test errors for a set of transfer learning experiments.

## 7 CONCLUSIONS

In this paper, we presented a novel approach called LaDID aimed at end-to-end learning of latent dynamical models. LaDID uses a novel transformer-based architecture that leverages certain scientifically-motivated invariances to allow separation of a universal dynamics module and encoded realization-specific information. We demonstated strong performance on several new and challenging test cases and well-known benchmarks. Additionally, we showed that LaDID can generalize to systems under nontrivial *intervention* (when trained on the un-intervened system) using few-shot learning. Currently, while LaDID accommodates irregular time sampling, data acquired from irregular *spatial* grids will need further work. A future research direction is to explore graph-based methodologies to address this specific challenge.

## ACKNOWLEDGEMENTS

This work was partly supported by the German Federal Ministry of Education and Research (BMBF) project "LODE", the UK Medical Research Council (MC-UU-00002/17) and the National Institute for Health Research (Cambridge Biomedical Research Centre at the Cambridge University Hospitals NHS Foundation Trust). The work of CL was funded by the Deutsche Forschungsgemeinschaft within the Walter Benjamin fellowship LA 5508/1-1. We gratefully acknowledge the Gauss Centre for Supercomputing e.V. for supporting this project via computing time on the GCS Supercomputers.

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

# A    EXTENDED RELATED WORK

We begin by discussing the wide variety of methods proposed to learn dynamical models from data. Our focus is on ML-based approaches.

**Operator learning and approximating dynamics.** Brunton et al. (2016); Kaheman et al. (2020) access Koopman theory to learn coefficients of a finite set of non-linear functions to approximate the observed dynamics. Other approaches (Cranmer et al., 2020; Lutter et al., 2019; Greydanus et al., 2019; Finzi et al., 2020; Zhong et al., 2020; Bai et al., 2019) induce Hamiltonian and Lagrangian priors into neural networks exploiting the reformulations of Newton's equations of motion in energy-conservative dynamics. This attractive inductive bias allows for accurate predictions over a significantly wider time-horizon but involves strong assumptions that may not hold for many real-world systems. Purely data-driven ML approaches have attracted much recent attention and have shown strong performance in many areas. For instance, Heinonen et al. (2018) learns a data-driven approximation of a dynamical system via a Gaussian Processes. Recent works focus on the field of operator learning that seeks to learn mappings between infinite-dimensional functions. Two prominent approaches are DeepONet (Lu et al., 2019) and Fourier Neural Operator (FNO; (Li et al., 2020)). DeepONet which is based on physics-informed neural networks (Raissi et al., 2019) can query any coordinate in the domain for a value of the output function. However, the input function must be observed on a set of predefined locations, requiring the same observation grid for all observations, for training and testing. FNO is an instance of neural operators, a family of approaches that integrate kernels over the spatial domain. Since this operation can be expensive, FNO addresses the problem by employing the fast Fourier transform (FFT) to transform the inputs into the spectral domain. As a consequence it cannot be used with irregular grids. Li et al. (2022) introduce an FNO extension to handle more flexible geometries, but it is tailored for design problems. To summarize, despite promising results for several applications, current operator approaches still face generalization limitations, and more importantly require access to the system's variable space to work sufficiently. As a result, the performance of these approaches drops dramatically if applied to longitudinal system representations and not the system itself.

**Incorporating strong physical priors in dynamics modeling.** In real-world dynamical systems it is often infeasible to have a reliable analytical model of the underlying processes. In such cases, a more general approach is to learn and capture the latent dynamics of the data using an architecture that incorporates an appropriate inductive bias. Hamiltonian and Lagrangian mechanics offer distinct mathematical reformulations of Newton's equations of motion, specifically for energy-conservative dynamics. The conservation of energy in these systems enables predictions of the system's state over significantly longer time horizons, both forward and backward, compared to the training data. This property makes them attractive biases to incorporate into deep neural networks. Despite the different coordinate frames they employ (Hamiltonian using position $q$ and momentum $p$, and Lagrangian using position $q$ and velocity $\dot{q}$), both formalisms describe the same underlying dynamics, allowing for seamless translation between the two without any loss of generality. One advantage of these models is that they only need to infer the Hamiltonian or the Lagrangian, without the additional burden of learning a state representation. This also simplifies evaluation, as it requires calculating the distance between the ground truth states and the states predicted by the model along the trajectory. To tackle this challenge, several approaches have been proposed that augment physics-inspired models with encoder/decoder modules Cranmer et al. (2020); Lutter et al. (2019); Greydanus et al. (2019); Finzi et al. (2020); Zhong et al. (2020); Bai et al. (2019); Wang et al. (2021); Zhang et al. (2023). These modules facilitate the inference of low-dimensional states from high-dimensional pixel observations bridging the gap between visual input and the underlying dynamics.

**Neural ODEs.** In a recent line of research, the ability of NNs to approximate arbitrary functions has been exploited for unknown dynamical models, with connections drawn between deep architectures and numerical solvers for ODEs, PDEs and SDEs (Chen et al., 2018; Weinan, 2017; Lu et al., 2018; Ruthotto and Haber, 2020; Haber and Ruthotto, 2017; Richter-Powell et al., 2022). This has given rise to new algorithms rooted in neural differential equations that have been shown to provide improved modeling of a variety of dynamical systems relative to standard recurrent neural networks (RNNs) and their variants. However, Neural ODEs inherit some key limitations of ODEs. Specifically, the temporal dynamics only depends on the current state but not on the history. This puts a theoretical limit on the complexity of the trajectories that ODEs can model, and leads to practical

consequences (e.g. ODE trajectories cannot intersect). Some existing works mitigate this issue by introducing latent variables (Rubanova et al., 2019) or higher order terms (Yildiz et al., 2019). Other attempts include combining recurrent neural nets with neural ODE dynamics (De Brouwer et al., 2019), where latent trajectories are updated upon observations, as well as upon Hamiltonian (Zhong et al., 2020), Lagrangian (Lutter et al., 2019), second-order (Yildiz et al., 2019), or graph neural network based dynamics (Poli et al., 2019), 2019]. Dupont et al. (2019) put forward a simple augmentation to the state space to broaden the range of dynamical systems that can be learned; Kidger et al. (2020) propose neural controlled differential equations; a mechanism to adjust the trajectory based on subsequent observation denoting the natural equivalent to recurrent neural networks, the work of Morrill et al. (2021) incorporate concepts of rough theory to extend applicability to trajectories spanning thousands of steps, and Jia and Benson (2019) explore extensions to account for interrupting stochastic events. However, they still operate in the ODE framework and cannot model broader classes of differential equations. Another limitation of Neural ODE and extensions is that they struggle to model certain types of dynamics due to numerical instability. This is because Neural ODE relies on a numerical ODE solver to predict the trajectory (forward pass) and to compute the network gradients (backward pass). Two common scenarios where standard numerical ODE solvers fail are (1) systems with piecewise external forcing or abrupt changes (i.e. discontinuities) and (2) stiff ODEs (Ghosh et al., 2020), both are common in engineering and biological systems. Some existing works address this limitation by using more powerful numerical solvers. Specifically, when modelling stiff systems, Neural ODE requires special treatment in its computation: either using a very small step size or a specialized IVP numerical solver (Kim et al., 2021a). Both lead to a substantial increase in computation cost.

**RNNs for modeling spatio-temporal dynamic systems.** RNNs, with their capability of capturing temporal dependencies in time sequences, have found extensive use in dynamic system modeling. A notable example is the data-driven forecasting method for high-dimensional chaotic systems using LSTM networks, as proposed by Vlachas et al. (2018). These networks can be used to infer high-dimensional dynamical systems in a reduced order space and have proven to be effective nonlinear approximators of their attractor. In a subsequent study,Vlachas et al. (2020) studied backpropagation algorithms and explored the merit of Reservoir Computing (RC) (Pathak et al. (2018)) and Backpropagation through time (BPTT) in RNNs for predicting complex spatio-temporal dynamics. The study revealed that RC outperforms BPTT in terms of predictive performance and capturing long-term statistics when the full state dynamics are available for training, but in the case of reduced order data, large scale RC models appear to be more unstable and tend to diverge faster compared to the BPTT algorithms. On the other hand, BPTT methods demonstrated superior forecasting performance in reduced order systems and better captured dynamics in these challenging conditions. In a different line of research, Koppe et al. (2019) proposed a state space model (SSM) leveraging generative piecewise-linear recurrent neural networks (PLRNN) to infer dynamics from neuroimaging data (fMRI). They outline the superior interpretability of the PLRNN model with respect to latent embeddings and showed that it is amenable to systematic dynamical systems analysis while allowing a straightforward transformation into an equivalent continuous-time dynamical system. In a follow-up study, the same group introduced a state space model based on a dynamically interpretable and mathematically tractable RLRNN augmented by a linear spline basis expansion (Brenner et al. (2022)). This approach retains all the theoretically appealing properties of the simple PLRNN while boosting its performance and accuracy when dealing with arbitrary nonlinear dynamical systems in comparatively low dimensions. That is, they empirically showed that the dendritically expanded PLRNN achieves better reconstructions with fewer parameters and dimensions on various dynamical systems benchmarks. In a similar vein, the same group of authors proposed a simple modification of the teacher forcing which allows for faithful reconstructions on noisy real-world data and yields an even more improved interpretability Hess et al. (2023). In complementary work, Schmidt et al. (2021) targeted a simpler regularization scheme for vanilla RNNs based on ReLU activation which aims to solve long-range dependency problems. Precisely, the proposed approach aims to express slow time scales while retaining a simple mathematical structure which makes their dynamical properties partly analytically accessible. Both proposed theorems demonstrate a strong link between the dynamics of regularized RNNs and their gradients, showcasing the effectiveness of their methodology across various benchmarks. Recently, Wang et al. (2022) presented PredRNN, a state-of-the-art recurrent network for predictive learning of spatiotemporal sequences that allows to model systematically visual appearances and temporal

dynamics of spatiotemporal observations. Based on the overall goal to generate future images by learning from the historical context, the authors proposed that visual dynamics have modular structures that can be learned with compositional subsystems using convolutional gated networks. In PredRNN, a pair of memory cells are explicitly decoupled to form unified representations of the complex environment and operate in nearly independent transition manners. Based on novel zigzag memory flow, this network enables the learning of visual dynamics at different levels of RNNs.

**Multiple shooting approaches.** Multiple shooting breaks up the time grid of an observed trajectory into some number of segments, and using single shooting method to solve for each segment. The idea behind this multiple shooting method stems from the observation that long integration of dynamics can be counterproductive for transcribing continuous optimal control problems, and this problem can be tackled by limiting the integration over arbitrarily short time intervals (Bock and Plitt, 1984). Hence, multiple shooting solves the ODE separately on each interval. In the context of neural ODE modeling, Turan and Jäschke (2021) proposed to combine multiple shooting with existing approaches drastically reducing the complexity of the loss landspape. That is, for long range time horizon optimal control or oscillation data driven problems, multiple shooting can be used to train the neural networks which helps to escape "flattened out" fitting for oscillation data. Concurrently, Massaroli et al. (2021) showed that multiple shooting can also be combined with neural differential equation and introduced differential multiple shooting layers. Iakovlev et al. (2023) extended this concept to sparse Bayesian multiple shooting, with both works evaluating latent NODEs.

**Attention-based time-series modeling.** The innovation of attention-based layers in deep learning (Vaswani et al., 2017) has been central in recent years in natural language processing, computer vision, and speech processing. Transformers have shown the ability to model long-range dependencies and interactions in sequential data and thus are relevant for time series modeling. Many variants of transformers have been proposed to address special challenges in time series modeling and have been successfully applied to various time series tasks, such as forecasting (Li et al., 2019; Zhou et al., 2022), anomaly detection (Xu et al., 2022; Tuli et al., 2022), and classification (Yang et al., 2021). In spatio-temporal forecasting, both temporal and spatiotemporal dependencies are taken into account in time series models for accurate forecasting. For instance in challenging computer vision tasks, such as traffic analysis and trajectory forecasting, Cai et al. (2020); Xu et al. (2020) propose an autoencoding structure that uses a self-attention module to capture temporal-temporal dependencies and a graph neural network module to model spatial dependencies. In Yu et al. (2020) an attention-based graph convolution mechanism is introduced that is able to learn a complicated temporal-spatial attention pattern to trajectory prediction. Targeting a different research topic, Earthformer (Gao et al., 2022) proposes a cuboid attention for efficient space-time modeling, which decomposes the data into cuboids and applies cuboid-level self-attention in parallel. It is shown that this framework achieves good performance in weather and climate forecasting. Recently, Liang et al. (2023) proposed a novel spatial self-attention module and a causal temporal self-attention module to efficiently capture spatial correlations and temporal dependencies, respectively. Furthermore, it enhances transformers with latent variables to capture data uncertainty. Multivariate and spatio-temporal time series are becoming increasingly common in applications, calling for additional techniques to handle high dimensionality, especially the ability to capture the underlying relationships among dimensions. Recently, several studies have demonstrated that the combination of GNN and attentions could bring significant performance improvements in multimodal forecasting (Li et al., 2021), enabling better understanding of the spatio-temporal dynamics and latent causality.

Our approach is inspired by this body of work in that we also use neural networks to learn latent dynamical models. However, two key differences are as follows. First, our models are specifically designed to separate input trajectories into a realization-specific and realization-invariant and hence, leverage a particular kind of scientifically-motivated inductive bias from the outset. Second, exploiting these invariances allows us to eschew explicit neural ODEs altogether, providing an arguably simpler, transformer-based scheme that can be trained in a straightforward fashion, but that, as we show, achieves excellent performance on unseen data from complex dynamical systems and that can even be leveraged for few-shot learning to generalize to nontrivial system interventions.

## B  DETAILS OF RISK-BASED FRAMING

In the main text we introduced the *sufficient encoding assumption* or SEA. Here we show how under SEA, the learning task can be framed in a particularly simple way. Note that in this section, we consider a simple, abstract version of the full problem, with the aim of understanding how the problem can be usefully placed in a learning framework. The notation in this section follows from Section 3.2 in the main text and is otherwise self-contained and intended to be readable. We emphasize that the development below does not map one-to-one with the architecture in Section 3.3 in the main text, but rather explains, at a high level, why such a scheme is at all possible in this context. The architecture in the main text can be regarded as one way to instantiate the general ideas below.

To fix ideas, we first consider the simple case with no latent space and then consider the latent dynamics case. Consider supervised training of a NN, with training (input, output) pairs of the form

$$\{\underbrace{(t, \hat{\theta}_r)}_{\text{input}}, \underbrace{x_t^{(r)}}_{\text{output}}\}_{(t,r) \in \text{Train}},$$

where, as in Section 3.2 in the main text, $\hat{\theta}_r$ is a candidate RS encoding, $t$ is a time point and $x_t^{(r)}$ the observation at time $t$ in realization $r$. The goal is to learn a mapping that takes the input (the RS encoding and query time point) and yields the correct system output, for any test pair $(t', r')$. Consider the prediction $\hat{x}_t^{(r)} = f(t; m(\hat{\theta}_r; \theta, \theta_m), \theta)$, and note that we can write the RHS as $h(t, \hat{\theta}_r; \Theta)$ where $\Theta = (\theta, \theta_m)$ is a RI parameter. This emphasizes the fact that the RHS is in fact a function (here $h$, itself involving combination of $f$ and $m$) of only the inputs $(t, \hat{\theta}_r)$, and therefore potentially learnable from training pairs. Note that the parameters of $h$ are entirely RI and hence the only RS information is carried by the encoding $\hat{\theta}_r$. It is easy to see that under SEA this construction provides the correct output by writing the RHS as: $f(t; m(\hat{\theta}_r; \theta, \theta_m), \theta) = f(t; \theta_r, \theta) = x_t^{(r)}$, i.e. the correct output. Thus, combining encoding $\hat{\theta}_r$ and function $h$ allows prediction of the time evolution of any realisation. In other words, even if the RS encoding is distant from the true RS parameter, under SEA there exists a RI function that can correct it, and the second NN aims to learn a function $h$ which combines these RI elements to provide the desired mapping.

Next, we briefly summarise arguments that extend the analysis to the case of latent dynamics. As in the main text, consider dynamics at the level of latent variables $z \in \mathbb{R}^q$ and again consider a model with RS and RI parts but at the level of the latents, i.e. $z_t^r = f(t; \theta_r, \theta)$. Recall that we assume the observables are given by (an unknown) function $g$ of the hidden state $z$.

We first consider the case in which the latent-to-observed mapping is RI, and then the more general case of a RS mapping.

**Case I: RI mapping.**  Assume the observable is given as $x_t^r = g(z_t^r; \theta_g)$, where $g : \mathbb{R}^q \to \mathbb{R}^p$ is the (true) observation process and $\theta_g$ is an RI parameter. Further, assume that we have an estimate $\hat{\theta}_r$ of the RS parameter that satisfies the sufficient encoding assumption (SEA). In a similar fashion, assume we have an estimate $\hat{\theta}_g$, which may be incorrect (in the sense of $\hat{\theta}_g \neq \theta_g$) but that satisfies:

$$\exists m_g, \exists \theta_{m_g} : \theta_g = m_g(\hat{\theta}_g; \theta, \theta_{m_g}) \tag{10}$$

That is, $\hat{\theta}_g$ admits an RI correction. As above, the correction is unknown and may potentially depend on true, RS parameters. Note also that subject to the existence of a correction the estimate (and implied mapping) may be potentially arbitrarily incorrect. In analogy to SEA, we call this the *sufficient mapping assumption* (SMA).

Now, consider training of a NN, with training (input, output) pairs of the form

$$\{(t, \hat{\theta}_r), x_t^r\}_{(t,r) \in Train} \tag{11}$$

We want to understand whether supervised learning of a model to predict output for arbitrary queries $(t, r)$ is possible. This is not obvious, since we now have training data only at the level of the observables, but the actual dynamics operate at the level of latents. Consider the following function $h_{SMA}$:

$$h_{SMA}(t, \hat{\theta}_r; \Theta) = g(f(t; m(\hat{\theta}_r; \theta, \theta_m); m_g(\hat{\theta}_g; \theta, \theta_{m_g})) \tag{12}$$

where $\Theta = (\theta, \theta_m, \theta_{m_g})$ is an RI parameter.

Under SEA and SMA it is easy to see that $h_{SMA}$ provides the correct output, since:

$$
\begin{aligned}
h_{SMA}(t, \hat{\theta}_r; \Theta) &= g(f(t; m(\hat{\theta}_r; \theta, \theta_m); m_g(\hat{\theta}_g; \theta, \theta_{m_g})) \\
&= g(f(t; \theta_r, \theta); \theta_g) \\
&= g(z_t^r; \theta_g) \\
&= x_t^r
\end{aligned}
$$

**Case II: RS mapping.** Suppose now the mapping is RS, with the model specification as above but with the observation step $x_t^r = g(z_t^r; \theta_g^r)$, where $\theta_g^r$ is an RS parameter. This means that the latent-observable relationship is itself non-constant and instead varies between realizations.

Assume we have a candidate estimate $\hat{\theta}_g^r$ which may be incorrect in the sense of $\hat{\theta}_g^r \neq \theta_g^r$ but that satisfies:

$$
\exists m_g, \exists \theta_{m_g} : \theta_g^r = m_g(\hat{\theta}_g^r; \theta, \theta_{m_g}) \tag{13}
$$

In analogy to SMA, we call this the realization-specific sufficient mapping assumption or RS-SMA. Now to create training sets, we extend the formulation to require input triples, as:

$$
\{(t, \hat{\theta}_r, \hat{\theta}_g^r), x_t^r\}_{(t,r) \in Train} \tag{14}
$$

Note that $\hat{\theta}_g^r$ in the input triples may be incorrect, but only needs to satisfy RS-SMA. As in Case I, we have training data only at the level of observables (not latents) but want to understand whether supervised learning of a model to predict output for arbitrary queries $(t, r)$ is possible. Consider the function $h_{RS-SMA}$:

$$
h_{RS-SMA}(t, \hat{\theta}_r, \hat{\theta}_g^r; \Theta) = g(f(t; m(\hat{\theta}_r; \theta, \theta_m), \theta); m_g(\hat{\theta}_g^r; \theta, \theta_{m_g})) \tag{15}
$$

where $\Theta = (\theta, \theta_m, \theta_{m_g})$ is an RI parameter. In a similar manner to Case I, it is easy to see that $h_{RS-SMA}$ provides the correct output under SEA and RS-SMA, since:

$$
\begin{aligned}
h_{RS-SMA}(t, \hat{\theta}_r, \hat{\theta}_g^r; \Theta) &= g(f(t; m(\hat{\theta}_r; \theta, \theta_m), \theta); m_g(\hat{\theta}_g^r; \theta, \theta_{m_g})) \\
&= g(f(t; \theta_r, \theta); \hat{\theta}_g^r) \\
&= g(z_t^r; \hat{\theta}_g^r) \\
&= x_t^r
\end{aligned}
$$

The simple arguments in the foregoing, based on an abstract version of the problem of interest, are intended to explain why (under the assumptions stated) it is possible to address the learning problem in a relatively straightforward manner. The key idea is that subject to the sufficient encoding/mapping assumptions there exists a function (conceptually a combination of various elements as above) that permits input-output mapping as required. Note that for successful prediction, it is not required to actually explicitly identify the elements of the combination (and nor do we attempt to do so). The architecture we propose in the main text represents one way to learn such a function.

## C MODEL, APPROXIMATE POSTERIOR AND ELBO

Here, we provide details about our model, the approximate posterior and the ELBO.

**Joint distribution.** Our graphical model consists of parameters $\Phi = \phi_{enc} \cup \phi_{dyn} \cup \phi_{dec}$, a random variable $\psi^r$ which additionally acts as global random variable at the level of latent states $z_{t_q}$ and observations $x_{t_q}$. The index $t_q$ refers to a specific queried time point within a trajectory. The joint distribution is given by

$$
p(x, z, \psi^r) = p(x, z|\psi^r)p(\psi^r) = p(x|z)p(z|\psi^r)p(\psi^r) \tag{16}
$$

with

$$
p(\psi^r) = \mathcal{N}(0, 1) \tag{17}
$$

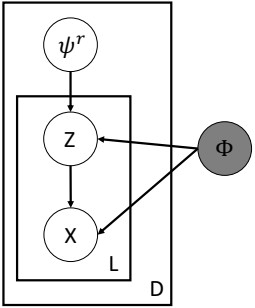

Figure A1: Graphical model: The graph consists of three random variables, the realization-specific trajectory representation $\psi^r$, the latent states $z$ and the observations $x$. Fixed parameters $\phi$ are represented by the grey node. Here, the plate notation $D$ indicates the independence between trajectories within the dataset, while the plate notation $L$ corresponds to the number of queried time points of the roll-out and denotes the conditional independence of predicted states given the trajectory representation $\psi^r$ and the RI parameters $\Phi$, i.e. $P(z_{t_{q_i}}|\psi^r, \Phi) \perp\!\!\!\perp P(z_{t_{q_j}}|\psi^r, \Phi); \; i \neq j$.

$$p(z|\psi^r) = \prod_{i=1}^{N} p(z_n|\psi_n^r)p(z_n|z_{n-1}, \psi_{n-1}^r) \tag{18}$$

$$p(x|z) = \mathcal{N}(f_{\phi_{dec}}(z), \sigma_{dec}). \tag{19}$$

**Approximate posterior.** Our graphical model assumes the following independencies: (i) The dataset contains i.i.d. trajectories of varying length. (ii) The observation of trajectory $x_{t_q}^r$ at time $t_q$ is conditionally independent of $x_{t_{q-1}}^r$ at time $t_{q-1}$, given latent states $z_{t_q}^r$ and trajectory representation $\psi^r$: $p(x_{t_q}|z_{t_q}, \psi^r) \perp\!\!\!\perp p(x_{t_{q-1}}|z_{t_{q-1}}, \psi^r)$. Analyzing data with this graphical model involves computing posterior distributions of hidden variables given observations

$$p(z, \psi^r|x) = \frac{p(x, z, \psi^r)}{\int p(x|z)p(z|\psi^r)p(\psi^r)dzd\psi^r}. \tag{20}$$

**ELBO.** To effectively manage long-horizon time series data, we adopt a variation of the multiple shooting approach. However, our model differs from those based on explicit ODE formulations, thus avoiding the need to transform initial value problems into boundary value problems (Massaroli et al., 2021). Instead, we incorporate a Bayesian continuity prior (Hegde et al., 2022; Iakovlev et al., 2023) to extend the multiple-shooting framework from deterministic neural ODEs to a probabilistic context. Our approach dissects each realization $x_{t:T}^r$ into a series of $N$ overlapping subtrajectories and independently condenses each patch into a latent representation. Within this Bayesian multiple shooting framework, the smoothness prior connects the patches via

$$p(z|\psi^r) = \prod_{i=1}^{N} p(z_n|\psi_n^r)p(z_n|z_{n-1}, \psi_{n-1}^r) \tag{21}$$

to form a cohesive global trajectory. We leverage the independence of trajectory representations in subpatches i.e. $p(z_i|\psi_i^r) \perp\!\!\!\perp p(z_j|\psi_j^r)$. For the continuity prior, we follow Hegde et al. (2022) and place a Gaussian prior on the error between consecutive subtrajectories, i.e. $\Delta \sim \mathcal{N}(0, \sigma_\Delta)$ entailing exact overlapping if $\Delta \to 0$. This yields our continuity prior

$$p(z_n|z_{n-1}, \psi_{n-1}^r) = \mathcal{N}((z_n^{t_1}|z_{n-1}^{-t}, \psi_{n-1}^r), \sigma_\Delta), \tag{22}$$

where the time index $-t$ refers to the last time point of a subpatch. The prior trajectory representation is set to a Gaussian, i.e. $p(\psi_n^r) \sim \mathcal{N}(0, 1)$.

ELBO:

$$
\begin{aligned}
\ln p(x) &= \ln \int p(x, z, \psi^r) \frac{q(z, \psi^r)}{q(z, \psi^r)} dz d\psi^r & (23) \\
&\geq \int q(z, \psi^r) \ln \frac{p(x, z, \psi^r)}{q(z, \psi^r)} dz d\psi^r & (24) \\
&= \int q(z, \psi^r) \ln \frac{p(x|z) p(z|\psi^r) p(\psi^r)}{q(z) q(\psi^r)} dz d\psi^r & (25) \\
&= \int q(z, \psi^r) \ln \frac{p(x|z) p(\psi^r) \prod_{n=1}^{N} p(z_n|\psi_n^r) p(z_n|z_{n-1}, \psi_{n-1}^r)}{q(z) q(\psi^r)} dz d\psi^r & (26) \\
&= \int q(z, \psi^r) \ln p(x|z) \prod_{n=1}^{N} p(z_n|\psi_n^r) dz d\psi^r + \int q(z, \psi^r) \ln \frac{p(\psi_n^r)}{q(\psi_n^r)} dz d\psi^r & (27) \\
&+ \int q(z, \psi^r) \ln \frac{p(z_1|z_0, \psi_0^r) \prod_{n=2}^{N} p(z_n|z_{n-1} \psi_{n-1}^r)}{q(z_n)} dz d\psi^r & (28)
\end{aligned}
$$

This is equivilant to maximizing eq. 9 of the manuscript with short term notation $p_n(\hat{x}_n) = p(x_n|z_n) p(z_n|\psi_n^r)$:

$$
\max \mathbb{E}_{q(z, \psi^r)} \sum_{n=1}^{N} \ln p(x_n|z_n) p(z_n|\psi_n^r) - \sum_{n=1}^{N} \mathrm{KL}(q(\psi_n^r)||p(\psi_n^r|x)) - \sum_{n=2}^{N} \mathbb{E}_{q(z, \psi^r)} \mathrm{KL}(q(z_n)||p(z_n|z_{n-1}, \psi_n^r))
$$
(29)

## D  ELBO COMPUTATION AND FORECASTING

To find the approximate posterior which minimizes the Kullback–Leibler divergence

$$
\mathrm{KL}[q(z_{1:N}, \psi_N^r)||p(z_{1:N}, \psi_N^r|x_{1:N})]
\tag{30}
$$

we maximize the ELBO which for our model is given via

$$
\max \mathbb{E}_{q(z, \psi^r)} \underbrace{\sum_{n=1}^{N} \ln p_n(\hat{x}_n)}_{\text{(i) likelihood}} - \underbrace{\sum_{n=1}^{N} \mathrm{KL}(q(\psi_n^r)||p(\psi_n^r|x_n))}_{\text{(ii) representation prior}} - \underbrace{\sum_{n=2}^{N} \mathbb{E}_{q(z, \psi^r)} \mathrm{KL}(q(z_n)||p(z_n|z_{n-1}, \psi_n^r))}_{\text{(iii) smoothness prior}}
\tag{31}
$$

The ELBO is computed as follows:

1. Encode $x_{t-k:t-1}^r$ to obtain $\rho_k^{TA}$
2. Sample latent embeddings via $l_k^{emb} \sim \mathcal{N}(f_\mu(\rho_k^{TA}), f_\sigma(\rho_k^{TA}))$
3. Compute latent trajectory $z_{1:N}$
4. Compute ELBO via data likelihood estimation, and KL divergence of representation and smoothness prior.

Since our approach leverage amortized variational inference, the reparameterization trick is applied when sampling $l_k^{emb}$. Expectations of the data likelihood are computed using Monte Carlo integration with one sample. KL divergence is computed in closed form.

**Forecasting - Inference.** Similar to training, also in forecasting a RS vector is derived for a specific test trajectory aggregating its first $K$ observations as detailed above. This latent representation is then evolved over time based on the learned dynamics model. Note that in contrast to NODE based

approaches, here we can query arbitrary points in time which in fact allows us to directly infer only the latent trajectory state of the query time steps. No evaluations on intermediate grid points are required saving substantial computational resources. Finally, inferred latent trajectory points are mapped back to the observation using the trained decoder.

# E   MODEL ARCHITECTURE

**Encoder.** The encoder is a collection of three NNs. First, features from the input observations $x^r_{t-k:t-1}$ are extracted using a convolutional neural network (CNN) parameterized by $\theta_{enc}$ and shared across all representations and patches. Specifically, the CNN encoder has the following architecture: three convolution layers (5x5 kernel, stride 2, padding 2) with batch norm and ReLU activations, one convolution layer (2x2 kernel, stride 2) with batch norm and ReLU activation. The channels of the respective CNN layers are doubled throughout. Finally, the downsampled image features are flattened and linearly projected to the output dimension. Hence, our encoder transforms the sequence of input observations to a sequence of feature vectors, $z^{(enc),r}_{t-k:T}=f_{\theta_{enc}}(x^r_{t-k:T})$.

Then, we compute the trajectory representation and the latent embedding as follows. Each input patch is split into two disjoint sets by time. The first $k \in K$ data points $\mathcal{M}_R=\{z^{(enc),r}_{t-k:t}\}$ are used to compute a trajectory specific representation distribution $\psi^r \sim q_{\Theta_R}(x^r_{t-k:t-1}) = \mathcal{N}(\mu_r, \sigma_r)$ and $\mu_r, \sigma_r = f_{\theta_R}(z^{(enc),r}_{t-k:t})$. In cases of irregularly sampled trajectories, we use a time threshold $\tau$ to define the representation set, $\mathcal{M}_R=\{z^{(enc),r}_{t_i}\}$, $t_i \in \{t<\tau\}$. We model $f_{\theta_R}$ as a transformer network with temporal attention. In other words, we consider the sequence feature vectors $z^{(enc),r}_{t-k:t}$ as an time-ordered sequence of tokens and transform each token according to the temporal distance to the other tokens. We compared two approaches of temporal attention (Iakovlev et al., 2023; Bulat et al., 2021) which performed roughly similar. First, temporal reweighting is performed as introduced in Iakovlev et al. (2023): $C_{ij} = c_{ij}/\sum_{k=1}^K c_{ik}$ with $c_{ij} = \exp(\langle W_Q h_i, W_K hj \rangle + \ln(\epsilon)(|t_j - t_i|/\delta)^p)$, where $\langle \cdot, \cdot \rangle$ denotes the dot product, $W_K$, $W_Q$, and $W_V$ represent the weight matrices for the query, key, value as in regular attention. $\epsilon$ and $\delta$ are constants. Hence, the larger the distance $|t_j - t_i|$ grows, the stronger the time-aware attention is reduced (Iakovlev et al., 2023). The parameter $\delta$ determines the distance threshold beyond which the scaling of regular attention occurs by at least $\epsilon$. Moreover, parameter $p$ governs the shape of the scaling curve. This methods works best for most of the dynamical systems. Second, we tested a temporal attention approach as defined in (Bulat et al., 2021). This time aware attention is given by $C^{TA}(t) = \sum_{t'=0}^{T-1} softmax(\frac{\langle W_Q \rho_t, W_K \rho_{t'}\rangle}{\sqrt{d}})W_V \rho_{t'}$.

Finally, the trajectory representation $\psi^r$ is obtained applying a *mean*-aggregation of the temporally transformed representation tokens.

**Dynamics model.** With initial density given by the encoder networks $q_{\Theta_L}(z_t|x^r_{t-k:T}, \psi^r)$, the density for all queried latent points (on a continuous time grid) can be predicted by $z^r_{t_q} \sim \mathcal{N}(\mu^r_{t_q}, \sigma^r_{t_q})$ with $\mu^r_{t_q}, \sigma^r_{t_q} = f_{\theta_{dyn}}(t_q, z_t, \psi^r)$. Note that this approach allows for latent state predictions at any time since the learned dynamics module $f_{\theta_{dyn}}$ is continuous in time and our variational model utilizes encoders only for obtaining the initial latent distribution. We also make use of the reparameterization trick to tackle uncertanties in both, the latent states and in the trajectory representations (Kingma and Welling, 2013). In our implementation $f_{\theta_{dyn}}$ consists of three linear layers, with the first two followed by a ReLU nonlinearity.

**Decoder.** The decoder maps the latent trajectory points back to the observational space. Hence, our implementation is fairly simple and comprises a set of transposed convolutional layers. In particular, it first projects latent trajectory points linearly followed by four transposed convolution layers (2x2 kernel, stride 2) with batch norm and ReLU nonlinearities. Finally, a convolutional layer (5x5 kernel, padding 2) with sigmoid function computes our output distribution. The channel dimension of the four transposed convolution layers is halved subsequently from layer to layer.

## F   TRAINING DETAILS AND HYPERPARAMETERS

Our implementation uses the PyTorch framework (Paszke et al., 2019). All modules are initialized from scratch using random weights. During training, an AdamW-Optimizer (Loshchilov and Hutter, 2017) is applied starting at an initial learning rate $\varepsilon_0 = 0.0003$. An exponential learning rate scheduler is applied showing the best results in the current study. Every network is trained for 30 000 epochs. At initialization, we start training at a subpatch length of 1 which is doubled every 3000 epochs. After the CNN encoder, 8 attention layers are stacked each using 4 attention heads. A relative Sin/Cos embedding is used as position encoding followed by a linear layer. The input resolution of the observational image data is $128 \times 128$ px. All computations are run on a single GPU node equipped with one Nvidia A100 (40 GB) and a global batch size of 16 is used. A full training run on the single pendulum, the double pendulum, the wave equation and the Navier-Stokes equation dataset requires approx. 14 h. A full training run on the reaction-diffusion system and the von Kármán vortex street requires approx. 8 h.

Table F.1: Training hyperparameters

| Hyperparameter | Value |
| --- | --- |
| LR schedule | Exp. decay |
| Initial LR | 3e-4 |
| Weight Decay | 0.01 |
| Global batch size | 16 |
| Parallel GPUs | 1 |
| Input resolution | $128 \times 128$ px |
| Number of input timesteps | 10 |
| Initial subpatch length | 1 |
| Number of epochs per subpatch length | 3000 |
| Latent dimension | 32 |
| attention mechanism | spatio-temporal |
| Number of attention blocks | 8 |
| Number of attention heads | 4 |
| Position Encoding | relative Sin/Cos encoding |

## G   DETAILS ON COMPARISON BASELINES

All baseline methods are similar to our proposed approach in the sense that they encode longitudinal observations in a latent space, simulate a low-dimensional latent trajectory and decode such to obtain future observations. Similar to our approach, NDP (Norcliffe et al., 2021) encodes observations into two latent variables (an "initial state" and a "global control of an ODE") and aggregates latent representations in a global representation via averaging. Dynamics are modeled using MLPs or convolutions and integrated over time using neural ODEs. The decoder outputs a Bernoulli distribution from which the prediction is sampled.

From a high-level perspective, MSVI (Iakovlev et al., 2023) works similar, but leverages a slightly modified encoder. Here, a transformer module is added while dynamics function and decoder are Bayesian MLPs and CNNs whose parameterisation is assumed to be Gaussian. Similar to our work, amortized multiple shooting is employed leveraging a smoothness prior. Hence, training is performed based on a sophisticated loss incorporating data, continuity, dynamics, and decoder priors.

Likewise, ODE2VAE (Yildiz et al., 2019) represents a variational inference framework based on Bayesian Neural Networks. Similar to our work, it encodes a number of observations into a latent initial state, which are explicitly shaped by a physics-motivated prior, e.g., the latent space is separated into a velocity and position part. Subsequently, these high-order dynamics are approximated by a BNN and are evolved over time. The decoder is similar to MSVI where both BNN priors are assumed to be Gaussian.

Similar to the other baselines, ODE-RNN (Rubanova et al., 2019) represents a family of time series models whose hidden state dynamics are specified by Neural ODEs. Hence, the trained ODE-RNN models rely on latent ODEs and can naturally handle arbitrary time gaps between observations. In this context, note that ODE-RNN is not applicable to spatio-temporal system dynamics by default. As a result, the same CNN encoding part is used as in our approach while the dynamics function and

decoder are similar to the ones used in MSVI. In this way, we aim to provide a useful comparison. All models are trained and tested according to default parameters and code provided in the original papers. Please note that for comparisons to these baseline methods, we are forced to rely on regular time grid datasets since ODE2VAE's encoder is exclusively applicable to evenly spaced grids. For irregularly sampled observations, we report comparisons to MSVI.

## H  INFERENCE AND TRAINING TIMES

Table H.2: Comparison of training and inference time as well as the number of trainable model parameters for all methods applied to the single pendulum test case. Note that for MSVI (block size = 1 / 8), no inference times can be given since inference requires roll-out across the full trajectory. All tests are performed on a NVIDIA A100 40 Gb with a AMD EPYC 7742 processor.

| | forward / backward pass [ms] | forward pass (inference) [ms] | trainable parameters [M params] |
|---|---|---|---|
| ODE-RNN | 851.95 | 351.68 | 10.51 |
| NDP | 388.69 | 163.15 | 1.13 |
| ODE2VAE | 571.77 | 68.04 | 3.04 |
| MSVI (block size = 60) | 1192.86 | 48.37 | 1.51 |
| MSVI (block size = 8) | 285.79 | (N/A) | 1.51 |
| MSVI (block size = 1) | 60.08 | (N/A) | 1.51 |
| LaDID (ours) | 48.11 | 15.09 | 1.35 |

## I  DATASET DETAILS

Here, we provide details about the datasets used in this work, their underlying mathematical formulations and their implementation details. Partially, some of the datasets we selected are used in literature to demonstrate the effectiveness of neural based temporal modeling approaches, e.g., a swinging pendulum or a reaction-diffusion system. However, we also consider more unknown test cases which represent complex real-world applications such as the chaotic double-pendulum or fluid flow applications driven by a complex set of PDEs, i.e., the Navier-Stokes equations.

### I.1  SWINGING PENDULUM

For the first dataset we consider synthetic videos of a nonlinear pendulum simulated in two spatial dimensions. Typically, a nonlinear swinging pendulum is governed by the following second order differential equation:

$$\frac{d^2 z}{dt^2} = -\sin z \tag{32}$$

with $z$ denoting the angle of the pendulum. Overall, we simulated 500 trajectories with different initial conditions. For each trajectory, the initial angle $z$ and its angular velocity $\frac{dz}{dt}$ is sampled uniformly from $z \sim \mathcal{U}(0, 2\pi)$ and $\frac{dz}{dt} \sim \mathcal{U}(-\pi/2, \pi/2)$. All trajectories are simulated for $t = 3$ seconds. The training, validation and test dataset is split into 400, 50 and 50 trajectories, respectively. The swinging pendulum is rendered in black/white image space over 128 pixels for each spatial dimension. Hence, each observation is a high-dimensional image representation (16384 dimensions - flatted $128 \times 128\,\mathrm{px}^2$ image) of an instantaneous state of the second-order ODE.

### I.2  SWINGING DOUBLE PENDULUM

To increase the complexity of the second dataset, we selected the kinematics of a nonlinear double pendulum motion. The pendulums are treated as two point masses with the upper pendulum being denoted by the subscript "1" and the lower one by subscript "2". The kinematics of this nearly chaotic

system is governed by the following set of ordinary differential equations:

$$\frac{d^2 z_1}{dt^2} = \frac{-g(2m_1 + m_2)\sin z_1 - m_2 g \sin(z_1 - 2z_2) - 2\sin(z_1 - z_2)m_2(\frac{dz_1}{dt}^2 L_2 + \frac{dz_1}{dt}^2 L_1 \cos(z_1 - z_2))}{L_1(2m_1 + m_2 - m_2 \cos(2z_1 - 2z_2))} \tag{33}$$

$$\frac{d^2 z_2}{dt^2} = \frac{2\sin(z_1 - z_2)(\frac{dz_1}{dt}^2 L_1(m_1 + m_2) + g(m_1 + m_2)\cos z_1 + \frac{dz_2}{dt}^2 L_2 m_2 \cos(z_1 - z_2))}{L_2(2m_1 + m_2 - m_2 \cos(2z_1 - 2z_2))} \tag{34}$$

with $m_i$ denoting the mass and the length of each pendulum respectively, and $g$ is the gravitational constant. Again, we simulated 500 trajectories split in sets of 400, 50 and 50 samples for training, validation and testing. The initial condition for $(z_1, z_2)$ and $(\frac{dz_1}{dt}, \frac{dz_2}{dt})$ are uniformly sampled in the range $\mathcal{U}(0, 2\pi)$ and $\mathcal{U}(-\pi/2, \pi/2)$. The double pendulum is rendered in a RGB color space over 128 pixels for each spatial dimension with the first pendulum colored in red and the second one in green. Hence, each observation is a high-dimensional image representation ($16384 \times 3$ dimensions - flatted $128 \times 128 \, \text{px}^2$ RGB image) of an instantaneous double pendulum state.

### I.3    REACTION-DIFFUSION EQUATION

Many real-world applications of interest originate from dynamics governed by partial differential equations with more complex interactions between spatial and temporal dynamics. One such set of PDEs we selected as test case is based on a lambda-omega reaction-diffusion system which is described by the following equations:

$$\frac{du}{dt} = (1 - (u^2 + v^2))u + \beta(u^2 + v^2)v + d_1(\frac{d^2 u}{dx^2} + \frac{d^2 u}{dy^2}) \tag{35}$$

$$\frac{dv}{dt} = -\beta(u^2 + v^2)u + (1 - (u^2 + v^2))v + d_2(\frac{d^2 v}{dx^2} + \frac{d^2 v}{dy^2}) \tag{36}$$

with $(d_1, d_2) = 0.1$ denoting diffusion constants and $\beta = 1$. This set of equations generates a spiral wave formation which can be approximated by two oscillating spiral modes. The system is simulated from a single initial condition from $t = 0$ to $t = 10$ in $\Delta t = 0.05$ time intervals for a total number of 10 000 samples. The initial conditions is defined as

$$u(x, y, 0) = \tanh\left(\sqrt{x^2 + y^2}\cos\left((x + iy) - \sqrt{x^2 - y^2}\right)\right) \tag{37}$$

$$v(x, y, 0) = \tanh\left(\sqrt{x^2 + y^2}\sin\left((x + iy) - \sqrt{x^2 - y^2}\right)\right). \tag{38}$$

The simulation is performed over a spatial domain of ($x \in [-10, 10]$ and $y \in [-10, 10]$ on grid with 128 points in each spatial dimension. We split this simulation into trajectories of 50 consecutive samples resulting in 200 in dependant realisations. We use 160 randomly sampled trajectories for training, 20 trajectories for validation and the remaining 20 trajectories for testing. Source code of the simulation can be found in Champion et al. (2019).

### I.4    TWO-DIMENSIONAL WAVE EQUATION

A classical example of a hyperbolic PDE is the two-dimensional wave equation describing the temporal and spatial propagation of waves such as sound or water waves. Wave equations are important for a variety of fields including acoustics, electromagnetics and fluid dynamics. In two dimensions, the wave equation can be described as follows:

$$\frac{\partial^2 u}{\partial t^2} = c^2 \nabla^2 u, \tag{39}$$

with $\nabla^2$ denoting the Laplacian operator in $\mathcal{R}^2$ and $c$ is a constant speed of the wave propagation. The initial displacement $u_0$ is a Gaussian function

$$u_0 = a \exp\left(-\frac{(x - b)^2}{2r^2}\right), \tag{40}$$

where the amplitude of the peak displacement $a$, the location of the peak displacement $b$ and the standard deviation $r$ are uniformly sampled from $a \sim \mathcal{U}(2,4)$, $b \sim \mathcal{U}(-1,1)$, and $r \sim \mathcal{U}(0.25, 0.30)$, respectively. Similar to Yin et al. (2023), the inital velocity gradient $\frac{\partial u}{\partial t}$ is set to zero. The wave simulations are performed over a spatial domain of ($x \in [-1,1]$ and $y \in [-1,1]$ on a grid with 128 points in each spatial dimension. Overall, 500 independent trajectories (individual initial conditions) are computed which are split in 400 randomly sampled trajectories for training, 50 trajectories for validation and the remaining 50 trajectories for testing.

## I.5    NAVIER-STOKES EQUATIONS

To ultimately test the performance of our model on complex real-world data, we simulated fluid flows governed by a complex set of partial differential equations called Navier-Stokes equations. Overall, two flow cases of different nature are considered, e.g., the temporal evolution of generic initial vorticity fields and the flow around an obstacle characterized by the formations of dominant vortex patterns also known as the von Kármán vortex street.
Due to the characteristics of the selected flow fields, we consider the incompressible two-dimensional Navier-Stokes equations given by

$$\frac{\partial u}{\partial t} + (u \cdot \nabla)u - \nu \nabla^2 u = -\frac{1}{\rho} \nabla p. \tag{41}$$

Here, $u$ denotes the velocity in two dimensions, $t$ and $p$ are the time and pressure, and $\nu$ is the kinematic viscosity. For the generic test case, we solve this set of PDEs in its vorticity form and chose initial conditions as described in Li et al. (2020). Simulations are performed over a spatial domain of ($x \in [-1,1]$ and $y \in [-1,1]$ on a grid with 128 points in each spatial dimension. Overall, 500 independent trajectories (individual initial vorticity fields) are computed which are split in 400 randomly sampled trajectories for training, 50 trajectories for validation and the remaining 50 trajectories for testing.

## I.6    FLOW AROUND A BLUNT BODY

The second fluid flow case mimics an engineering inspired applications and captures the flow around a blunt cylinder body, also known as von Kármán vortex street. von Kármán vortices manifest in a repeating pattern of swirling vortices caused by the unsteady flow separation around blunt bodies and occur when the inertial forces in a flow are significantly greater than the viscous forces. A large dynamic viscosity of a fluid suppresses vortices, whereas a higher density, velocity, and larger size of the flowed object provide for more dynamics and a less ordered flow pattern. If the factors that increase the inertial forces are put in relation to the viscosity, a dimensionless measure - the Reynolds number - is obtained that can be used to characterize a flow regime. If the Reynolds number is larger than $Re > 80$, the two vortices in the wake of the body become unstable until they finally detach periodically. The detached vortices remain stable for a while until they slowly dissociate again in the flow due to friction, and finally disappear. The incompressible vortex street is simulated using an open-source Lattice-Boltzmann solver due to computational efficiency. The governing equation is the Boltzmann equation with the simplified right-hand side (RHS) Bhatnagar-Gross-Krook (BGK) collision term (Bhatnagar et al., 1954):

$$\frac{\partial f}{\partial t} + \zeta_k \frac{\partial f}{\partial x_k} = -\frac{1}{\tau}(f - f^{eq}) \tag{42}$$

These particle probability density functions (PPDFs) $f = f(\vec{x}, \vec{\zeta}, t)$ describe the probability to find a fluid particle around a location $\vec{x}$ with a particle velocity $\vec{\zeta}$ at time $t$ (Benzi et al., 1992). The left-hand side (LHS) describes the evolution of fluid particles in space and time, while the RHS describes the collision of particles. The collision process is governed by the relaxation parameter $1/\tau$ with the relaxation time $\tau$ to reach the Maxwellian equilibrium state $f^{eq}$. The discretized form of equation 42 yield the lattice-BGK equation

$$f_k(\vec{x} + \zeta_k \Delta t, t + \Delta t) = f_k(\vec{x}, t) - \frac{1}{\tau}(f_k(\vec{x}, t) - f_k^{eq}(\vec{x}, t)). \tag{43}$$

The standard $D_2Q_9$ discretization scheme with nine PPDFs (Qian et al., 1992) is applied. The equilibrium PPDF is given by

$$f_k^{eq} = w_k \rho \left( 1 + \frac{\zeta_k \vec{u}}{c_s^2} + \frac{(\zeta_k \vec{u})^2}{2c_s^4} - \frac{\vec{u}^2}{2c_s^2} \right) \tag{44}$$

where the quantities $w_k$ are weighting factors for the $D_2Q_9$ scheme given by $4/9$ for $k \in 0$, $1/9$ for $k \in 1, \ldots, 4$, and $1/36$ for $k \in 5, \ldots 9$, and $\vec{u}$ is the fluid velocity. $c_s$ denotes the speed of sound. The makroscopic variables can be obtained from the moments of the PPDFs. Within the continuum limit, i.e., for small Knudsen numbers, the Navier-Stokes equations can directly be derived from the Boltzmann equation and the BGK model (Hänel, 2006). We simulated three different Reynolds numbers $Re = 100, 250, 500$ for 425 000 iterations with a mesh size of 128 point in vertical and 256 points in horizontal direction. We skipped the first 25 000 iterations to ensure a developed flow field and extracted velocity snapshot every 100 iterations. The simulation is performed over a spatial domain of ($x \in [-20, 20]$ and $y \in [-10, 10]$). We split this simulation into trajectories of 50 consecutive samples resulting in 200 in dependant realisations. We use 160 randomly sampled trajectories for training, 20 trajectories for validation and the remaining 20 trajectories for testing.

## J    DETAILED DESCRIPTION OF EXPERIMENTS

We structured our experiments into four different series that shed light on the performance of LaDID in different situations with unique contextual information and challenges.

1. **ODE based systems:** Here, we investigate the performance for simulated kinematic systems. In these experiments, the right-hand side of the ODE remains constant, so the underlying dynamical system is the same for training and test cases. However, every trajectory starts from a unique initial conditions that lead to unique trajectories in the observational space (image space). The set of initial conditions of the training data and the test data is entirely disjoint. We tested two kinematic systems:

    (a) Single Pendulum: We simulated 400 training trajectories at an image resolution of $128 \times 128$ pixels and tested on 50 distinct test trajectories. The dataset generation is described in Subsection I.1. The results are presented in Figure 3, 4, and K.1.

    (b) Double Pendulum: We simulated 400 training trajectories at an image resolution of $128 \times 128$ pixels and tested on 50 distinct test trajectories. The dataset generation is described in Subsection I.2. The results are presented in Figure K.2.

2. **PDE based systems:** The series of experiments focuses on the performance of LaDID in PDE based systems. Similar, to the experimental setup of the ODE experiments, the dynamical system described by the right-hand side of the PDE remains constant throughout this set of experiments. Every trajectory corresponds to the temporal evolution under the dynamical system given a unique initial condition. Here, we focus on four different PDE based dynamical systems:

    (a) Reaction-Diffusion Equation: The dataset contains 160 training trajectories and 20 distinct test trajectories at an image resolution of $128 \times 128$ pixels. The generation of the datasets is given in Subsection I.3 of the Appendix and the results are shown in Figure K.3.

    (b) Wave Equation: The dataset comprises 400 training trajectories and 50 unique test trajectories, all at an image resolution of $128 \times 128$ pixels. Details on how the datasets were generated are provided in Subsection I.4 of the Appendix, and the corresponding results are depicted in Figure K.4.

    (c) Navier-Stokes Equations: The dataset comprises 400 training trajectories and 50 unique test trajectories, each with an image resolution of $128 \times 128$ pixels. Detailed information on the dataset generation is provided in Subsection I.5 of the Appendix, with corresponding results displayed in Figure K.5.

    (d) Flow around a blunt body: We generated 160 training trajectories with an image resolution of $128 \times 128$ pixels and conducted testing on 20 unique test trajectories. The details of dataset generation can be found in Subsection I.6. The corresponding results are illustrated in Figure 3, 4, and K.1.

3. **Regular vs. Irregular time grid:** Here, we delve into the impact of time grid regularity on performance. We systematically compare the model's predictions when trained on datasets featuring a regular time grid against those with an irregular time grid. The objective is to study how the model adapts to variations in the temporal structure of the input data and assess its generalization capabilities across different time grid configurations. These experiments aim to provide insights into robustness and flexibility in handling diverse time representations, shedding light on suitability for real-world scenarios with irregularly sampled temporal data. We study two test cases:

   (a) Single Pendulum
   (b) Reaction-Diffusion Equation

   The regular case is identical to the experiments for the ODE/PDE systems as described above. For cases with irregular time grid, we uniformly sample the time points along the time axis and compute the corresponding observations in the image space. The description of the datasets is detailed in the Subsection I.1 and I.3, respectively. The results for the single pendulum test is shown in Figure L.1 and in Figure L.2 for the reaction-diffusion equation.

4. **Transfer Learning:**
   *Few-shot learning and zero-shot learning under interventions upon the dynamical system*
   In this set of experiments, we focus on evaluating transferability to new dynamical systems, emphasizing scenarios where the target systems share similarities with the training distribution but have not been encountered during the training phase. We test this in two different ways. First, we adopt a few-shot learning paradigm, wherein a pretrained model is fine-tuned using a limited number of trajectories from the new system. This set-up allows us to assess the model's adaptability to novel dynamical systems and to examine its capacity to generalize knowledge from the training set to previously unseen systems. To study transferability, we used the single pendulum dataset as described in Subsection I.1 of the Appendix and fine-tuned the model on a dataset that comprises $n_{new} = [16, 32, 64, 128]$ fine-tuning trajectories (which corresponds to $p = [4\%, 8\%, 16\%, 32\%]$ of the training dataset size) obtained through unique interventions on the mass and length of the pendulum. Additionally, all initial conditions of the trajectories are unique. We conducted testing on 50 unique test trajectories that were subject to new interventions and never seen during fine-tuning. Note that interventions here yield dynamical systems that are qualitatively different from the others and in that sense unique. In other words, the right-hand side of the underlying dynamical system is different for all interventions. The results are presented in Section 6.2 and Figure 6a of the main text and Section P in the Appendix.

   Furthermore, we extend our investigation to encompass zero-shot learning, wherein our approach is initially trained on a broader class of dynamical systems. Subsequently, we assess the model's ability to predict entirely new dynamical systems without additional training on specific instances. These experiments aim to elucidate ability to leverage learned knowledge to make accurate predictions for diverse dynamical systems. We chose the flow around a blunt cylinder as our test case and generated a training dataset that comprises 120 trajectories for three different Reynold's number $Re = [100, 250, 500]$. We test zero-shot transferability for three new Reynold's numbers $Re_{new} = [175, 300, 400]$. The results are presented in Subsection O and Figures P.1 and P.2 of the Appendix.

   *Interventions upon the measurement process of the dynamical system*
   In this series of experiments, we explore the robustness and adaptability of our approach under interventions that mimic variations in the measurement process of the dynamical system. Specifically, we examine the impact of additive Gaussian noise and salt & pepper noise, aiming to understand performance under observational variability. Additionally, we systematically alter channels in the RGB space to investigate the model's response to changes in color representations, providing insights into its ability to handle variations in the visual input domain. These effects are studied using the single-pendulum test case. The results are presented in Subsection M.1 for perturbed input images using Gaussian additive and salt & pepper noise. Subsection M.2 shows the effects of alteration in the RGB channels.

Furthermore, we extend our analysis to include experiments involving few-shot learning, where the model is presented with trajectories captured from entirely new camera angles and positions. This design enables us to assess the model's capacity to adapt and generalize to novel observational conditions with minimal exposure to the new data. These experiments collectively provide information on resilience to diverse sources of measurement noise and environmental changes. This is relevant for real-world applications with varying observational conditions. In this particular experiment, we selected the flow around a blunt cylinder as our test case. Utilizing a pretrained model with a fixed camera position, we proceeded to fine-tune it using a dataset comprising $n_{new} = [6, 12, 24, 48]$ distinctive trajectories captured from various camera positions. This corresponds to subsets representing $p = [4\%, 8\%, 16\%, 32\%]$ of the training dataset size, providing a systematic exploration of model adaptation to diverse camera perspectives. Testing is carried on an entirely unseen dataset with $n = 20$ test trajectories under new camera positions. The results are presented in Section 6.2 and Figure 6b of the main manuscript.

# K ADDITIONAL RESULTS

## K.1 SWINGING SINGLE PENDULUM

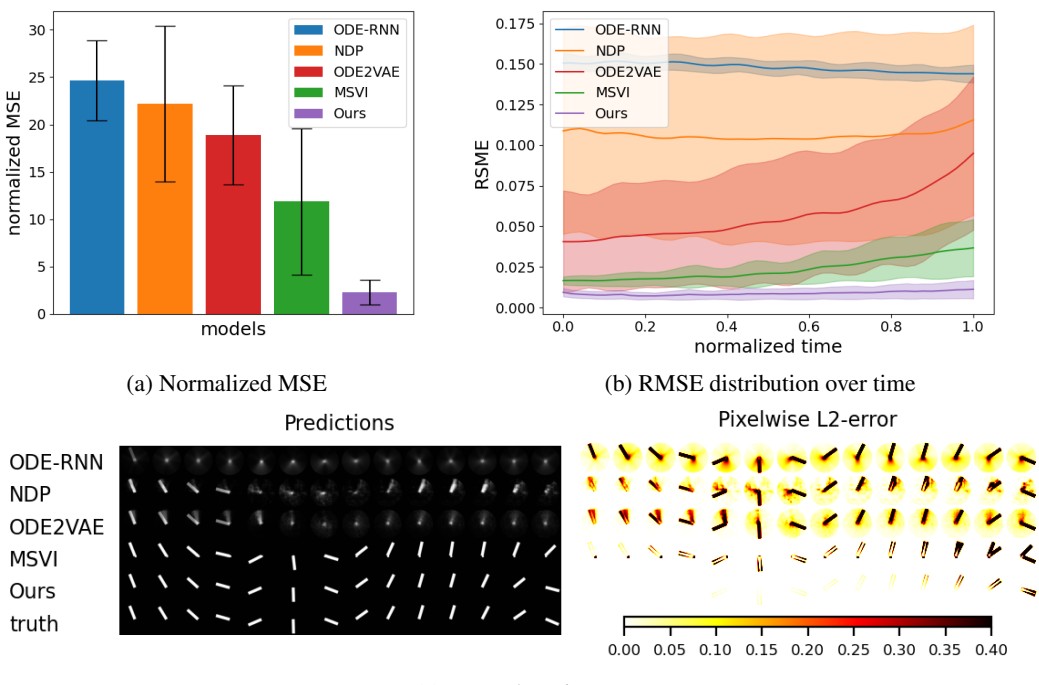

(a) Normalized MSE

(b) RMSE distribution over time

(c) Example trajectory

Figure K.1: Test errors and exemplary test trajectories of different models for the single pendulum test case.

## K.2 SWINGING DOUBLE PENDULUM

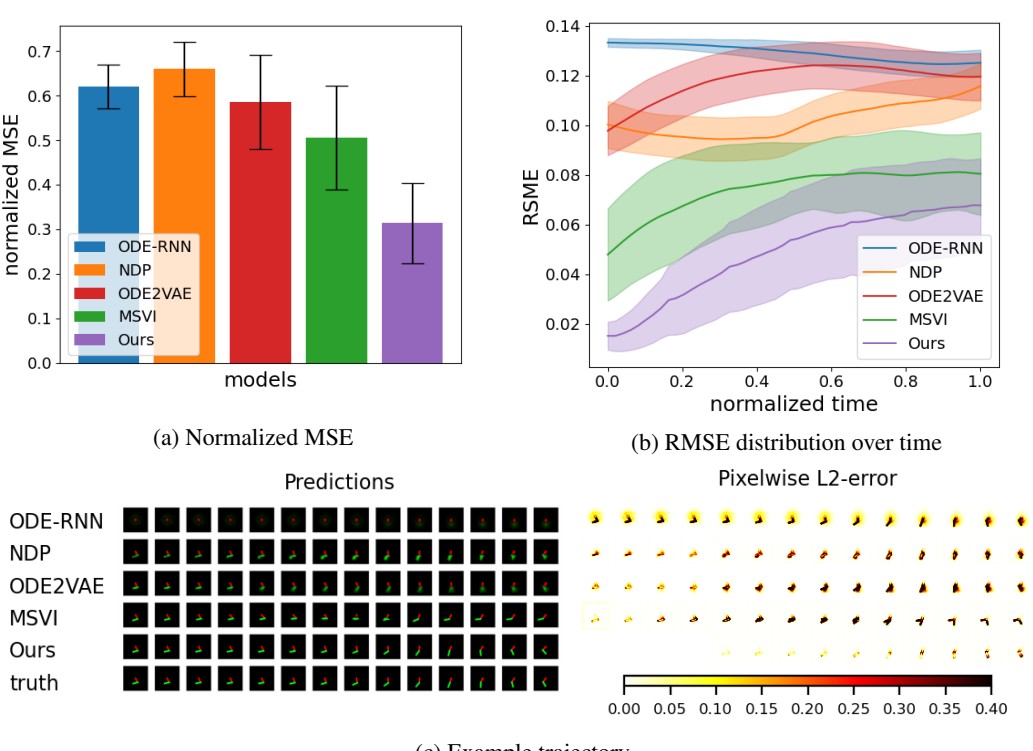

(a) Normalized MSE

(b) RMSE distribution over time

(c) Example trajectory

Figure K.2: Test errors and exemplary test trajectories of different models for the double pendulum test case.

## K.3 LAMBDA-OMEGA REACTION-DIFFUSION SYSTEM

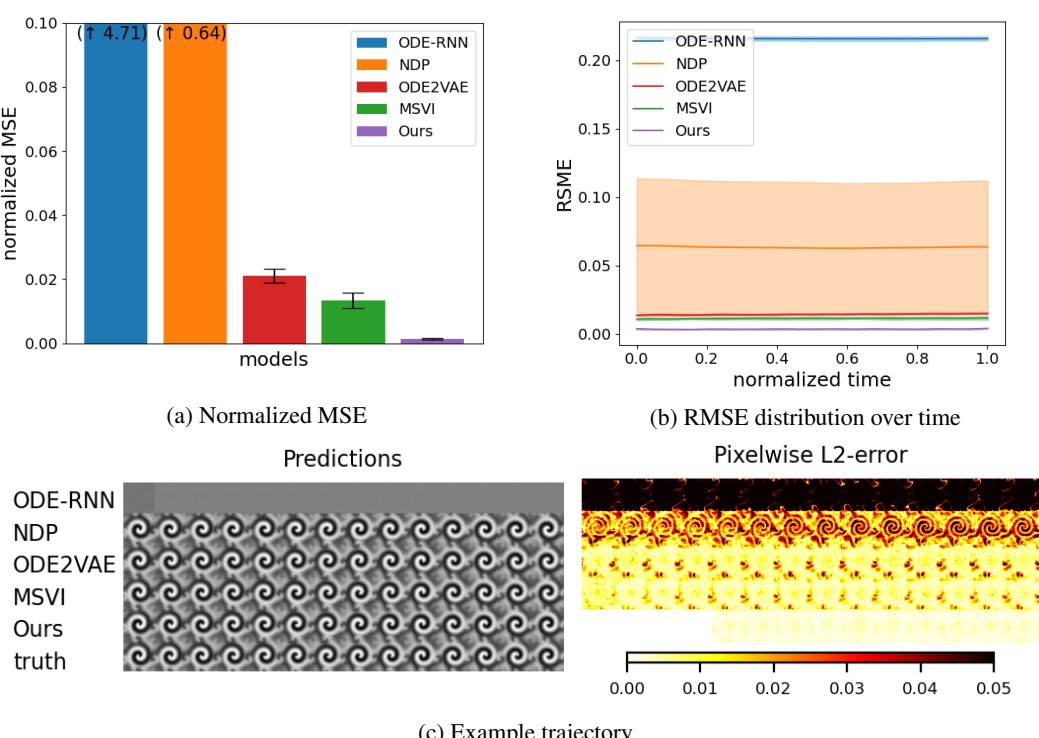

(a) Normalized MSE

(b) RMSE distribution over time

(c) Example trajectory

Figure K.3: Test errors and exemplary test trajectories of different models for the lambda-omega reaction-diffusion system.

## K.4   Two-dimensional wave equation

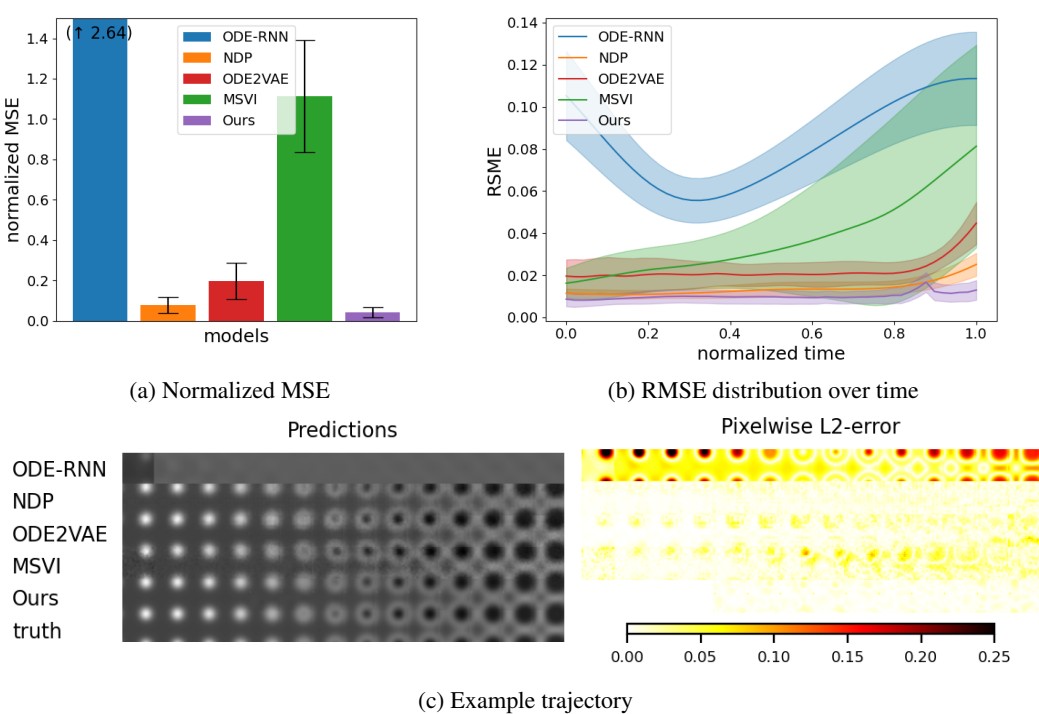

(a) Normalized MSE

(b) RMSE distribution over time

(c) Example trajectory

Figure K.4: Test errors and exemplary test trajectories of different models for the wave equation test case.

## K.5 NAVIER-STOKES EQUATIONS

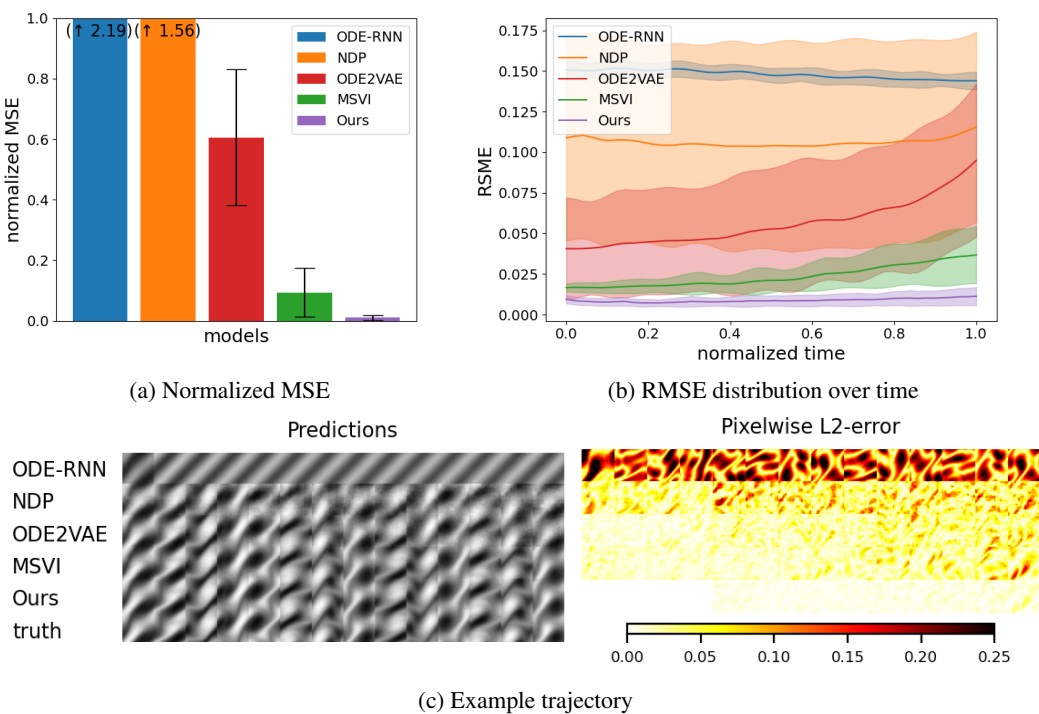

(a) Normalized MSE

(b) RMSE distribution over time

(c) Example trajectory

Figure K.5: Test errors and exemplary test trajectories of different models for the Navier-Stokes equations test case.

### K.6  LATTICE BOLTZMANN EQUATIONS

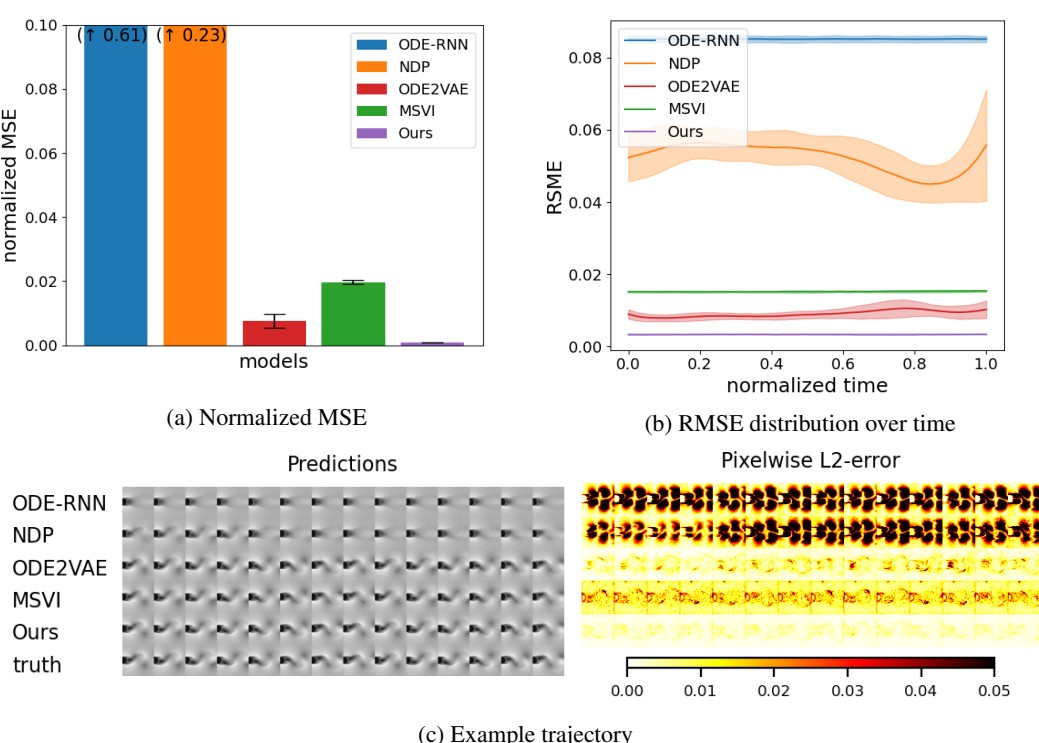

(a) Normalized MSE

(b) RMSE distribution over time

(c) Example trajectory

Figure K.6: Test errors and exemplary test trajectories of different models for the von Kármán vortex street test case.

## K.7 RESTRICTED ACCESS TO TRAINING TRAJECTORIES

In this experiment, we explore the impact of limiting the number of trajectories available during training. Specifically, we assess the performance of LaDID when utilizing only 4%, 8%, 16%, and 32% of the single pendulum dataset, which originally consists of 400 training trajectories. Please see Section I.1 for details on the data generation. In each experiment, our model is trained from scratch with a restricted number of trajectories. Comparisons are drawn against baseline algorithms which are trained on the full dataset. The results indicate that even with just 8% of the data, the model achieves reasonable predictions. Further, at 16%, LaDID surpasses the performance of all baseline algorithms.

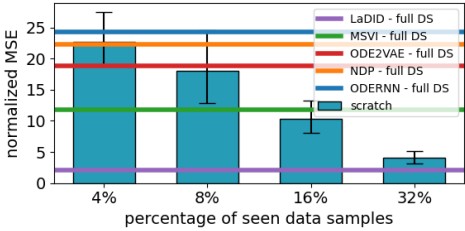

Figure K.7: Comparison of prediction performance of LaDID as the number of training trajectories varies. The model's accuracy is evaluated using different percentages of the standard dataset, ranging from 4% to 32%. Full dataset performances are represented by horizontal lines for LaDID and all baselines.

## K.8 INVESTIGATING REALIZATION-SPECIFIC AND REALIZATION-INVARIANT INFORMATION

LaDID is rooted in the idea of RS/RI decomposition, as reflected in both model architecture and the theory. In this section we provide an empirical study to investigate the RS/RI aspect. In particular, to better understand model intermediates, we carried out an experiment tracking the learned trajectory representation $\psi^r$ over a set of pendulum configurations (single pendulum) for which we know the true initial conditions $\pi_r$ of the underlying ODE system. According to the theory, it should be possible to recover the latter from the former. To test this, we use a trained LaDID model and compute the RS trajectory representations $\psi^r$ for every initial condition of the training dataset. In the next step, we learn a simple mapping function $g$ that maps the trajectory representations $\psi^r$ to the initial conditions $\pi_r$, i.e. $g : \mathbb{R}^{32} \to \mathbb{R}^2$ ($g$ is chosen to be a shallow 2 layer MLP with approximately 500 parameters). Training and testing are performed on two mutually disjoint sets of trajectory representations. That is, for training we computed the trajectory representations $\psi^r_{train}$ from the original *training dataset* using the initially trained LaDID encoder in a 1-step multiple shooting setting. This means that we split an observed trajectory into multiple subtrajectories and compute for each of these subtrajectories an independent $\pi_r$. For testing, we compute similar trajectory representations $\psi^r_{test}$, but this time from the *test dataset* ensuring that these are entirely disjoint from the training dataset since neither the LaDID encoder nor the learned mapping have seen these samples before. At testing, with mapping function $g$ and LaDID model fixed, we then predict the ground truth initial conditions, i.e. $\hat{\pi}_r = g(\psi^r)$. We present the results as scatter plots in Figure K.8; the good correspondence supports the notion that $\psi^r$ indeed captures the RS information. In addition, to understand whether the computed trajectory representation of any point of a trajectory holds information on the corresponding ground truth initial condition (as in ODE/PDEs), we compute for each time step of a trajectory its corresponding independent trajectory representation and predict the $\pi_r$'s as outlined before. The results are presented in Figs. K.8c & K.8d. The x-axis denotes the time, i.e. the time point from which we would start forecasting the dynamical behaviour, while the y-axis presents the predicted and ground truth initial conditions, i.e. angle $\Theta$ and $\hat{\Theta}$ and angular

velocity $\dot{\Theta}$ and $\hat{\Theta}$ respectively (note that markers shown in the figures are separate experiments with independent trajectory representations/initial conditions).

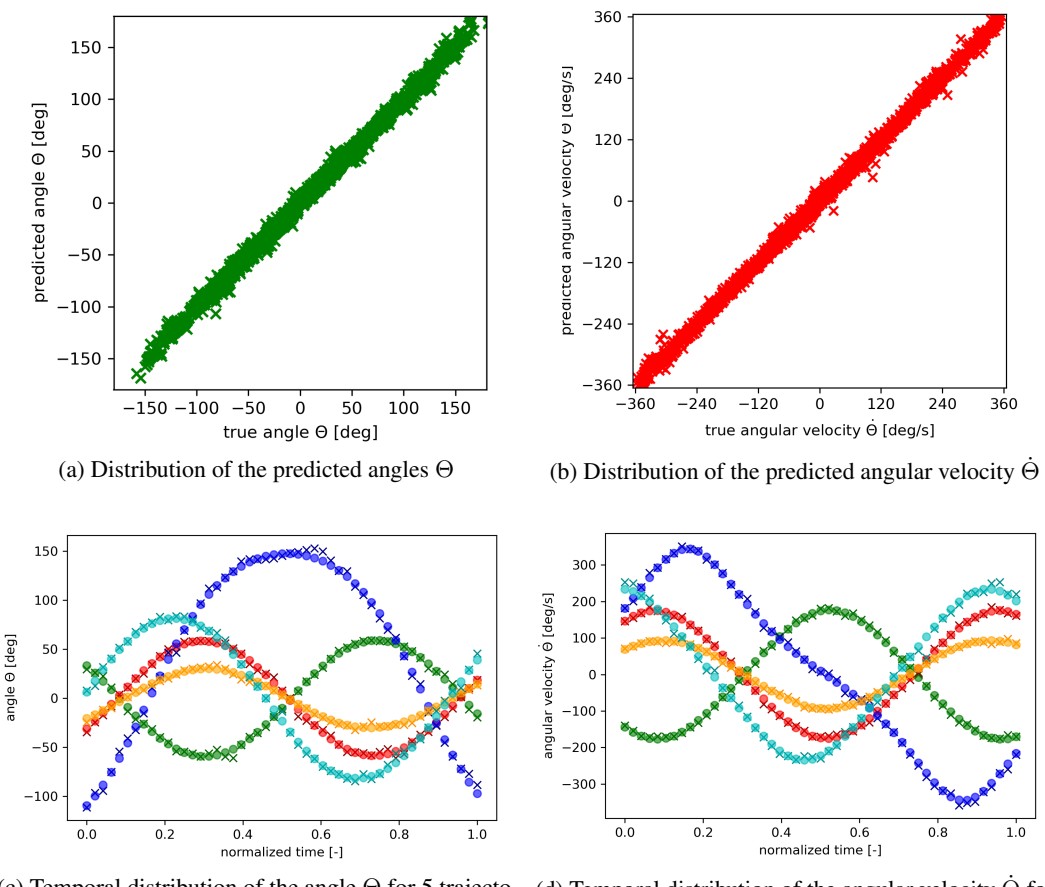

(a) Distribution of the predicted angles $\Theta$

(b) Distribution of the predicted angular velocity $\dot{\Theta}$

(c) Temporal distribution of the angle $\Theta$ for 5 trajectories

(d) Temporal distribution of the angular velocity $\dot{\Theta}$ for 5 trajectories

Figure K.8: Comparison of the recovered (predicted nonlinear mapping function) and true initial conditions of the swinging pendulum dataset: (a) Predicted vs true angles $\Theta$ of the entire test dataset, (b) Predicted vs true angular $\dot{\Theta}$ of the entire test dataset. (c) Temporal distribution of the angles $\Theta$ for five different trajectories. Note that the each marker corresponds to an initial trajectory representation $\psi^r$. (d) Temporal distribution of the angular velocity $\dot{\Theta}$ for five different trajectories.

## L  REGULAR AND IRREGULAR TIME GRIDS

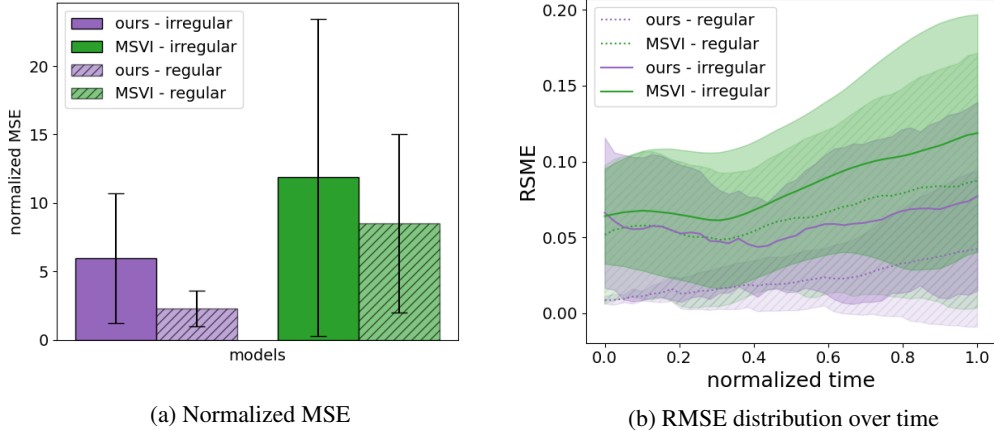

(a) Normalized MSE
(b) RMSE distribution over time

Figure L.1: Test errors and exemplary test trajectories of different models for regular and irregular time grids of the single pendulum test case.

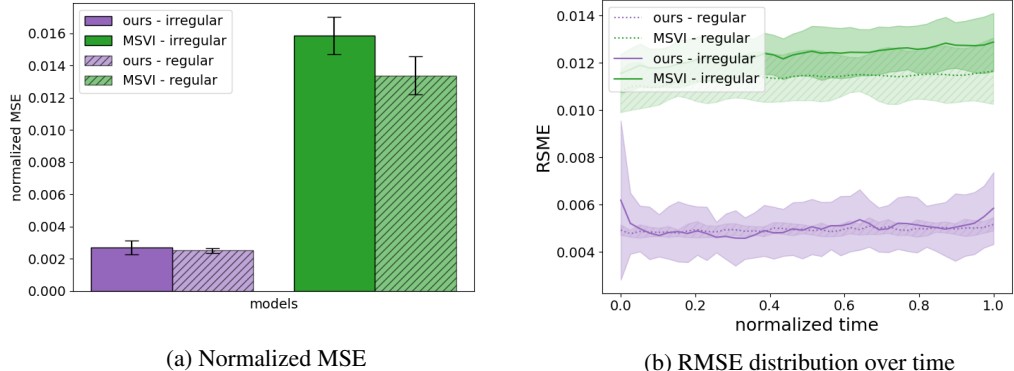

(a) Normalized MSE
(b) RMSE distribution over time

Figure L.2: Test errors and exemplary test trajectories of different models for regular and irregular time grids of the lambda-omega reaction-diffusion system.

## M  ROBUSTNESS TO INPUT NOISE AND APPEARANCE MODIFICATION

### M.1  ROBUSTNESS TO INPUT NOISE

In the following experiments, we study the impact of zero-mean Gaussian noise and salt-and-pepper noise on performance. Specifically, we vary the standard deviation for Gaussian noise ($\sigma = [0.1, 0.2, 0.3, 0.4]$) and the density for salt-and-pepper noise ($\rho = [10\%, 35\%, 50\%, 70\%]$) to systematically assess robustness. To provide a visual representation, we show example trajectories that exhibit the effects of these noisy inputs.

Next, we evaluate the summed MSE over trajectories, averaged across the test dataset. Additionally, we introduce the RMSE distribution plotted over time as detailed in Section 5 of the main text, offering insights into the model's response to noise over the entire trajectory duration. These results demonstrate robustness against the tested sources of noise for LaDID. We see minimal impact for small noise levels, with an expected increase in error values as noise intensity increases.

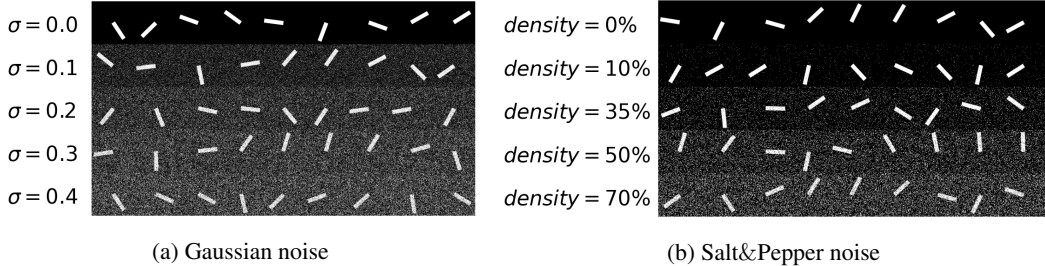

(a) Gaussian noise            (b) Salt&Pepper noise

Figure M.1: Visualization of test dataset augmentations to test the robustness to input noise.

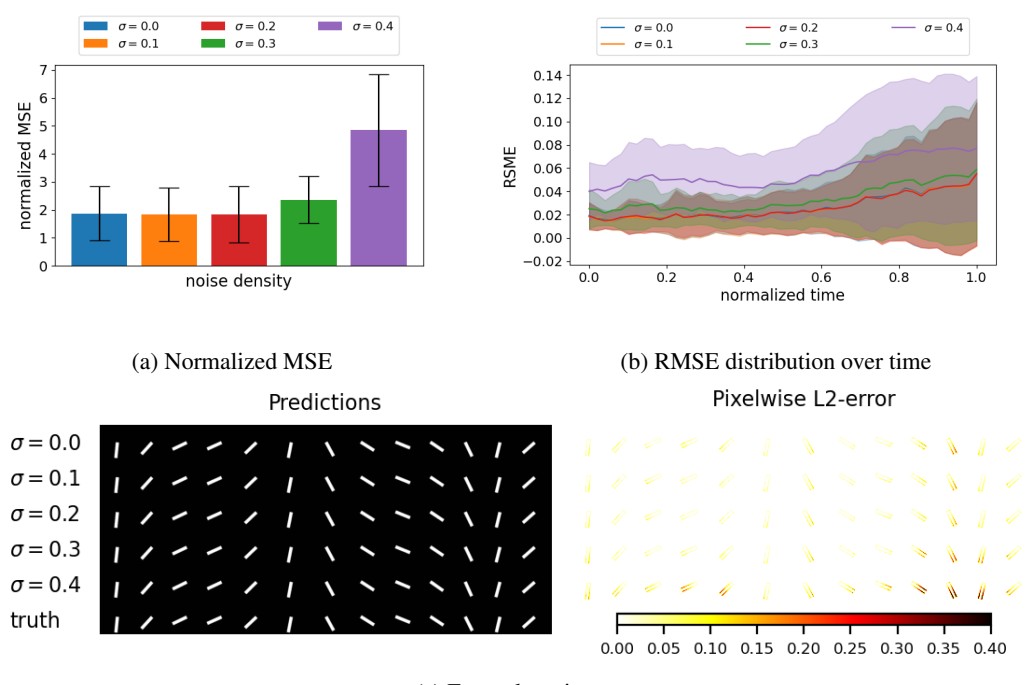

(a) Normalized MSE          (b) RMSE distribution over time

(c) Example trajectory

Figure M.2: Test errors and exemplary test trajectories of the single pendulum test case (gray-scale) corrupted by Gaussian Noise with various noise levels.

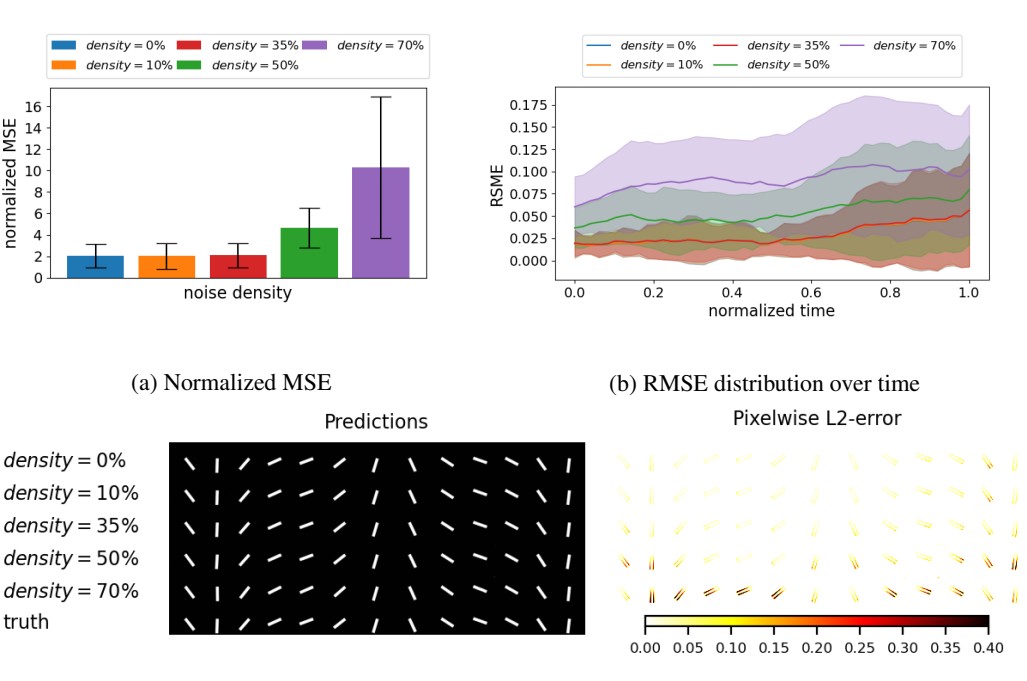

(a) Normalized MSE

(b) RMSE distribution over time

(c) Example trajectory

Figure M.3: Test errors and exemplary test trajectories of the single pendulum test case (gray-scale) corrupted by Salt&Pepper noise with various noise levels.

## M.2    ROBUSTNESS TO MODIFICATION IN THE INPUT APPEARANCE

We explore the impact of random scaling and flipping applied to the RGB channels of input images. This investigation aims to assess how the model responds to such spatial transformations. To provide a visual understanding, we start by showing example trajectories, allowing an examination of the perturbed inputs resulting from these transformations.

Following the visual inspection, we present a quantitative analysis, presenting the summed MSE over trajectories and averaged across the test dataset. Additionally, we introduce the RMSE distribution plotted over time, providing insights into the model's performance over the trajectory duration. The results indicate only slightly higher error values under the influence of these spatial transformations.

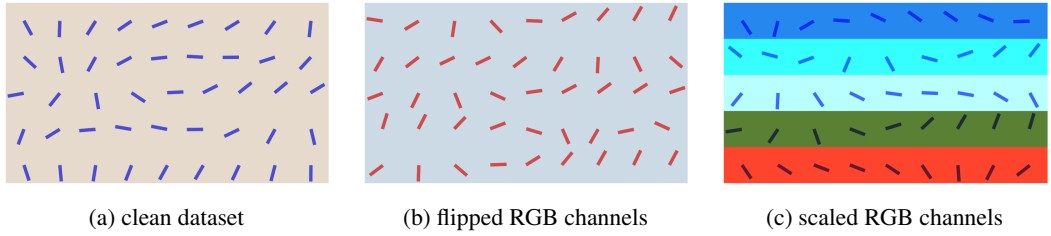

| (a) clean dataset | (b) flipped RGB channels | (c) scaled RGB channels |

Figure M.4: Visualization of test dataset augmentations to test the robustness to modifications of the input appearance.

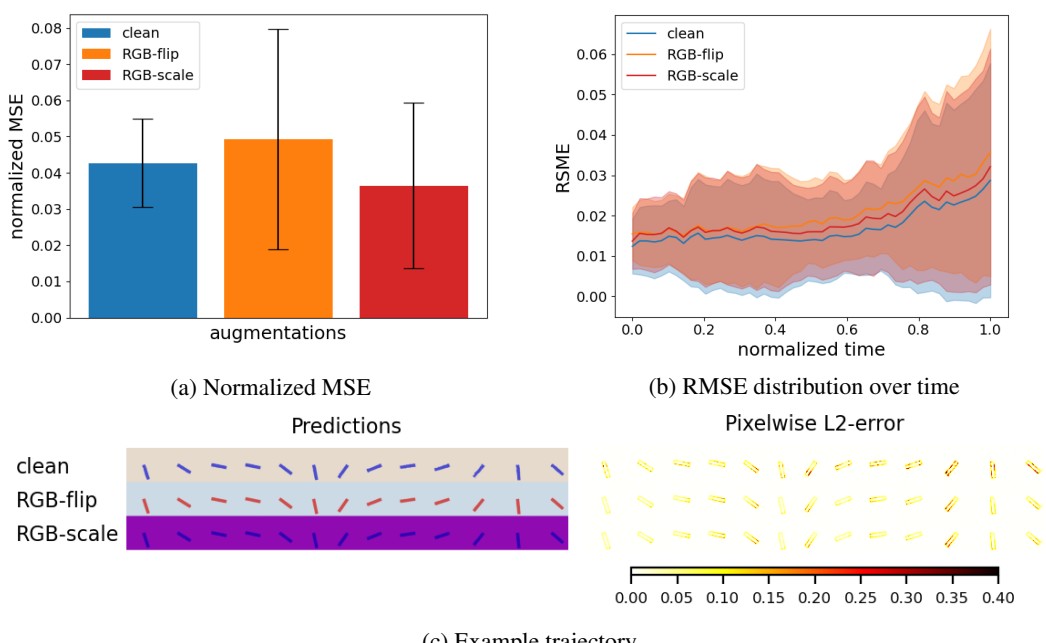

(a) Normalized MSE    (b) RMSE distribution over time

(c) Example trajectory

Figure M.5: Test errors and exemplary test trajectories of different RGB single pendulum test cases transformed by color augmentations.

## N  ABLATIONS - EFFECTS OF RELEVANT NETWORK MODULES

Table N.3: Errors for ablated LaDID models trained on the single pendulum test case.

| loss heuristics | | | attention mechanism | | | representation encoding | | |
|---|---|---|---|---|---|---|---|---|
| ablation | mean | IQR | ablation | mean | IQR | ablation | mean | IQR |
| reconstruction | 2.66 | 1.04 | no attention | 7.79 | 6.84 | w./o. encoding | 2.83 | 2.02 |
| reconstruction & representation | 2.17 | 0.99 | spatial attention | 2.81 | 1.47 | **w. encoding** | **2.02** | **0.88** |
| reconstruction & smoothness | 2.04 | 0.92 | temporal attention | 2.41 | 0.92 | | | |
| **full loss** | **2.02** | **0.88** | **spatio-temporal attention** | **2.02** | **0.88** | | | |

## O  GENERALIZATION UNDER INTERVENTIONS

We evaluated our model's ability to generalize with respect to aspects of the data generating distribution that are often unknown. Consider vectorized differential equation of the form

$$\frac{d}{dt}\boldsymbol{x} = f(\boldsymbol{x}, t) \tag{45}$$

with general initial conditions $\boldsymbol{x}(t_0) = \boldsymbol{x}_0$. Using an appropriate discretization scheme allows us to foretell the future of the system based on its past states and we can directly read off the causal structure for such a system. This underlying causal structure remains the same when changing coefficients on the right hand side which is summarized by the principle of independent causal mechanisms (Peters et al., 2017; Schoelkopf et al., 2012; Schölkopf et al., 2021; Lagemann et al., 2023b):

*Independent Causal Mechanism:* Let $X$ be the outcome variable, and let $M_1, M_2, \ldots, M_n$ be $n$ distinct mechanisms or factors that independently contribute to $X$. Then, we can write:

$$X = f(M_1, M_2, \ldots, M_n) \tag{46}$$

where $f$ represents the functional relationship between the mechanisms and the outcome. In other words, the outcome variable $X$ is determined by the independent contributions of each mechanism $M_i$, and these mechanisms operate independently of each other and do not inform each other.

For example, consider a system of ODEs that describes the dynamics of a population of predator and prey animals:

$$\frac{dx}{dt} = ax - bxy \tag{47}$$

$$\frac{dy}{dt} = -cy + dxy \tag{48}$$

where $x$ represents the population of prey, $y$ represents the population of predators, and $a$, $b$, $c$, $d$ are parameters that describe the interactions between the two populations. In this system, $x$ and $y$ represent independent causal mechanisms that contribute to the dynamics of the population. The equations describe how the population of prey and predators changes over time as a result of their interactions. By solving the system of ODEs, we can study how changes in the parameters affect the long-term behavior of the system, and how interventions can be used to control the population dynamics. Mathematically, an intervention is typically represented as a modification of the equations that describe the system, by setting the value of one or more variables to a fixed value or function. This modification represents the assumption that the variable(s) being intervened upon is no longer subject to external influences, and its value is determined by the intervention. In formal terms, an intervention upon a set of nodes in the causal structure of a system $\{X_i : i \in \mathcal{I}\}$ means any manipulation of the system that alters its state or behaviour, including changes in the initial conditions, modifying the parameters, or adding or removing variables or equations. When observing a latent process, which moves the deterministic setup of an ODE to a probabilistic case, this means that the

conditional distribution when observing the state of a node given its parents $Pa(X_i)$ is replaced by a new, predefined distribution. Thus, the joint probability distribution of as system under an intervention changes to

$$\tilde{p}(X) = \prod_{i \notin \mathcal{I}} p(X_i | Pa(X_i)) \prod_{i \in \mathcal{I}} \hat{p}(X_i | Pa(X_i)) \tag{49}$$

where $\hat{p}(X_i | Pa(X_i))$ indicates the conditional distribution of node $X_i$ in its general form.

## P    ADDITIONAL TRANSFER LEARNING RESULTS

In this set of experiments, our focus turns to the robustness and generalization capabilities in the face of novel fluid flow scenarios. Initially trained on a dataset comprising trajectories of fluid flows around a blunt object at Reynolds numbers $Re = [100, 250, 500]$, we consider zero-shot predictions on qualitatively different flows with Reynolds numbers $Re_{new} = [175, 300, 400]$. Without additional training on these new systems, we assess adaptability and accuracy in capturing the dynamics of unseen scenarios.

To evaluate performance, we present the summed MSE averaged across the test set, providing an overall measure of prediction accuracy. Additionally, we introduce the RMSE distribution over time, offering insights into the temporal dynamics of prediction errors.

Furthermore, we include a comparative analysis between the ground truth and predicted trajectories for each Reynolds number, offering a visual representation of the model's predictive capabilities in these novel conditions. As anticipated, we observe an increase in error for systems with unseen Reynolds numbers. However, even in these challenging scenarios, LaDID shows reasonable predictions in the observation space. These results underscore LaDID's adaptability and potential for generalization.

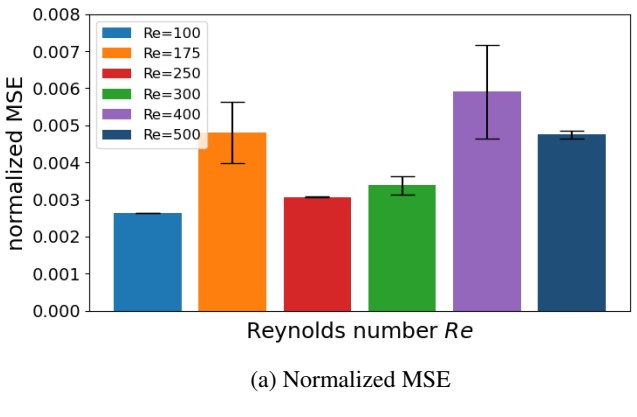

(a) Normalized MSE

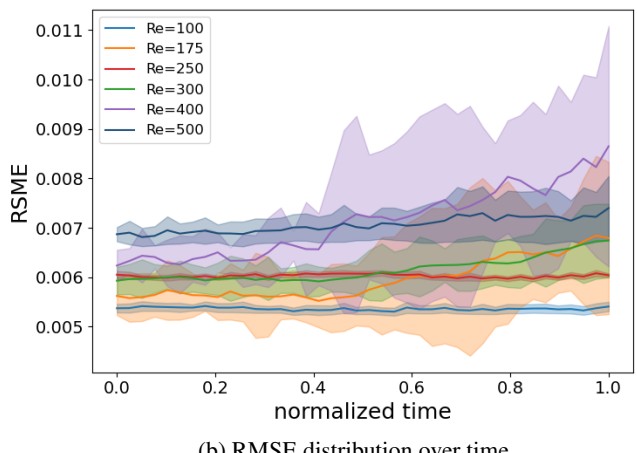

(b) RMSE distribution over time

Figure P.1: Transfer Learning: Test errors of various vortex street test cases characterized by a changing Reynolds number.

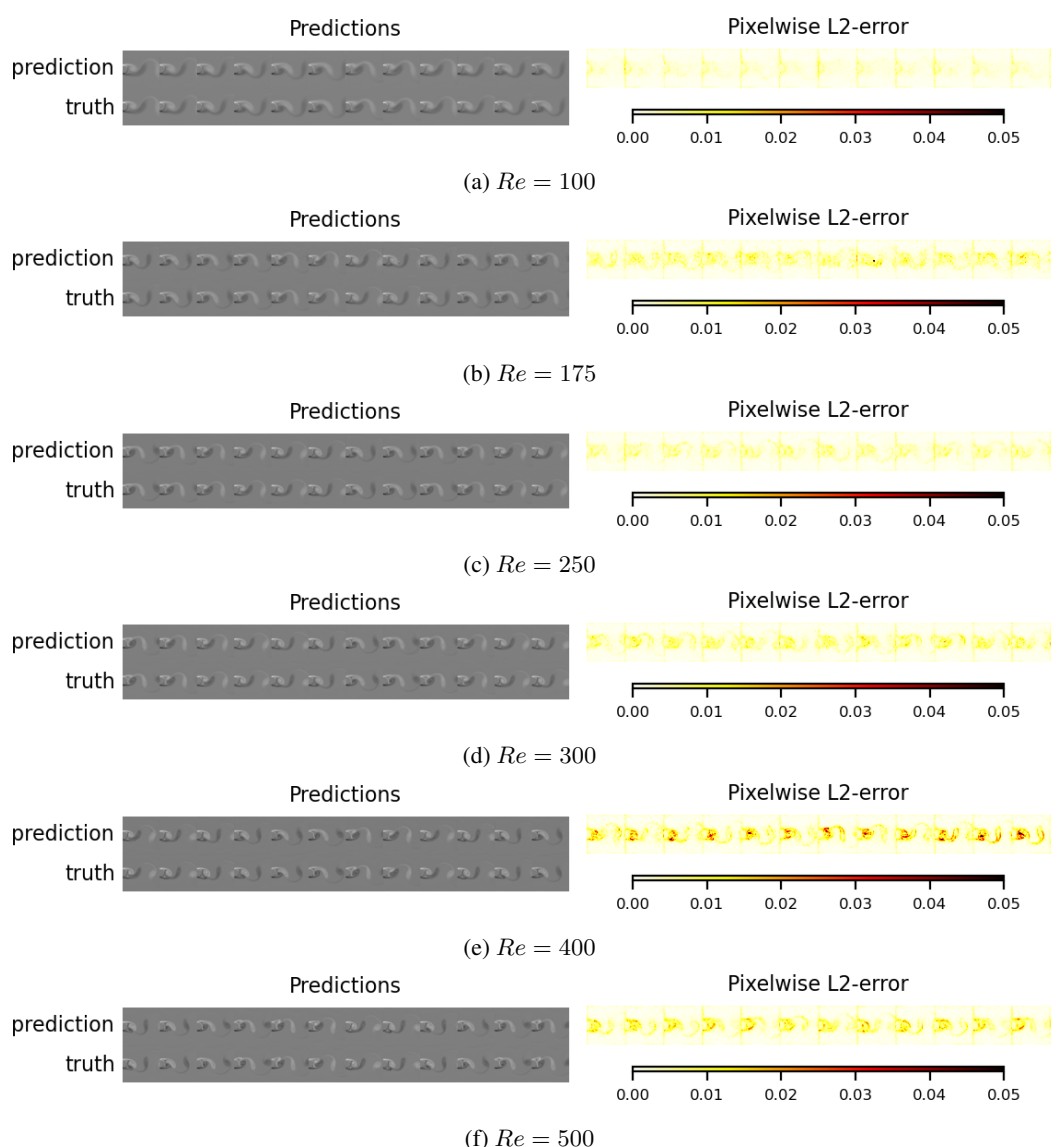

Figure P.2: Transfer Learning: Visualization of test trajectories of different vortex street test cases characterized by a changing Reynolds number.

## Q    LICENSES BASELINES

- MSVI (Iakovlev et al., 2023): MIT license
- ODE2VAE (Yildiz et al., 2019): The license status is unclear.
- NDP (Norcliffe et al., 2021): MIT license
- ODE-RNN (Rubanova et al., 2019): MIT license

