# OpenReview forum: "Invariance-based Learning of Latent Dynamics"
_ICLR.cc/2024/Conference — ICLR 2024 poster_

### Official Review · Reviewer_Fm8i · 2023-10-26

**Soundness:** 2 fair
**Presentation:** 3 good
**Contribution:** 2 fair
**Rating:** 5
**Confidence:** 3

**Summary:**

The paper introduces a novel framework called "Latent Dynamics via Invariant Decomposition" (LaDID) for predicting dynamical trajectories from high-dimensional empirical data. LaDID merges variational autoencoders with spatio-temporal attention, emphasizing scientifically-motivated invariances, allowing the model to predict system behavior at any continuous time and generalize beyond seen training data. Using a transformer-based architecture, it distinguishes between system-specific and universal dynamics, showing efficiency and scalability in experiments. The method is validated on various spatio-temporal systems, LaDID outperforms current neural-dynamical models.

**Strengths:**

1. The paper uniquely combines variational autoencoders with spatio-temporal attention

2. empirical tests on various spatio-temporal systems demonstrate that LaDID not only has practical utility but also massively outperforms existing neural-dynamical models

**Weaknesses:**

1.The definition of a system with realization-specific and realization-invariant properties is not clear. In chapter 3, I understand the authors try to explain its difference to the ode like systems and want to extend the ODE-like system setting to a more general settings with RS and RI properties, but I cannot find a specific example which is included in the RS-RI setting but excluded in the ODE setting. Can the author provide a useful example?

It seems like all the experiments in the paper can be treated as ODE-like dynamical system, so I was wondering what is the major difference here?

2.There is a missing literature review regarding invariant learning and its application in data-driven dynamical system learning, there are a few branches of related work worth mentioning and they are very close to the topic in this paper:
Physics/Conservation invariant:

"Hamiltonian neural networks." Advances in neural information processing systems 32 (2019).

Lagrangian Neural Networks. In ICLR 2020 Workshop on Integration of Deep Neural Models and Differential Equations.

ConCerNet: A Contrastive Learning Based Framework for Automated Conservation Law Discovery and Trustworthy Dynamical System Prediction. International Conference on Machine Learning 2023.

Symmetry invariant: Incorporating Symmetry into Deep Dynamics Models for Improved Generalization, ICLR 2021

Actually the Hamiltonian Neural network paper provides an image-based pendulum example which is similar to the experiment in this paper.

3.Regarding the novelty of the model structure, I feel like it still belongs to the encoder-latent dynamics-decoder model class like many other prior works. Despite the author claimed the model decomposes the realization-specific (RS) and realization-invariant (RI) information, there is no evidence proving the claim both experimentally and theoretically.

4.Honestly I’m a bit skeptical regarding the massive improvement of the proposed method over the existing work, especially in figure 4 and 5 (see my questions)

**Questions:**

1. In the performance comparison figure 4, is the X axis the time axis? The results surprised me because the prediction from the first 3 models is bad starting from the beginning of the prediction, which means the error mostly comes from decoder reconstruction rather than dynamics prediction. Especially for ODE2VAE, VAE should be able to reconstruct the pendulum image fairly well.

2. In Figure 5, the MSE from the proposed method is 10-100 times smaller than the prior works. This is another surprise as such a difference usually happens between white box model and black box model. Could there be bugs in your code, or is there any unfair comparison? I’m not trying to be mean but it does not look reasonable for me unless formal claims/evidence is provided.

3. In the latent dynamics model (part v in figure 1), how does the query time t work? Does this model need to solve it iteratively like ODE-like solver to time t or the neural network directly takes t as input and outputs the latent state at time t.

---

> ### Author Response · Authors · 2023-11-17
>
> Q1: Definition of RI and RS properties / Real-world motivation\
> Thank you for this question. Indeed, as you point out, our approach shares similarities with (neural) ODE approaches, but seeks to provide much more flexibility to bypass limits on the complexity of the trajectories that ODEs can model. Please see the overall response for a detailed description of the novelty of our contribution.
>
> Our long-term real-world motivation comes from modelling of trajectories in the biomedical domain:
> Here, underlying processes are too complex to permit explicit description, but we think the invariances we identify can still hold.  Based on our experience (neural) ODE approaches are typically not well suited to capture these complex, high-dimensional systems including strong system delays, discontinuities and distribution shifts (e.g. under  treatment), and therefore require the development of more general and flexible dynamics models.
>
> Q2: Extension of related work\
> We added a new subsection to Section A - Extended Related Work in the Appendix according to your suggestion and included the mentioned work, amongst others. Thank you for this suggestion.
>
> Q3: Contribution of work / Guarantee of decomposition\
> Thank you for this feedback, but we politely disagree with the reviewers opinion. Please see the overall response for a precise description of the fundamental differences of our method to other approaches and discussion re: decomposition in RI dynamical model and RS trajectory representation.
>
> Of course, we agree that specific model components like convolutions, attention, and VAEs are not new, but combining these ideas under our theoretical view and the presented graphical model results in a novel and demonstrably powerful inference approach.
>
> Q4: Code release to prove empirical performance\
> We are sorry that you are not convinced.  To provide more  confidence in the empirical work, we have released an anonymous code repository as outlined in the global answer.
>
> Q5: Poor baseline performance at the start\
> Thank you for mentioning this detail. You are right, the x-axis depicts the temporal evolution of the system. Regarding your concern, there is a simple explanation: In Fig. 4, the visualization of the predicted trajectory starts after $25\\%$ of the entire length meaning that only future time predictions are visualized without the initial reconstruction of the input. Unfortunately, we missed to provide this detail in the figure caption, but amended it now for clarification. Apologies for the inconvenience.
>
> Q6: Analogy to black/white-box model / masssive emperical performance boost\
> This is a great observation and your analogy of black and white-box model describes perfectly the difference between our approach and neural ODE based systems. Based on the conditional independence between latent states as outlined in the global answer we can directly predict the latent state $z_{t_{t_q}}$ without needing any information from previous states.  As you correctly point out,  this is equivalent to a black-box model for state transitions. In contrast, other approaches rely on the Markov assumption, meaning that they are modeling explicitly the transition between consecutive states. This is effectively a white-box description of state transition which we mitigate with our novel approach.
>
> For all baseline methods we used the code provided in the respective official repositories. Therefore, we do not think it is likely that bugs underlie the results.  Moreover, the test results in Appendix J including Figs. J.1 - J.6 nicely illustrate that the baselines are indeed able to reconstruct and predict future trajectory samples, but simply not at the same level of accuracy compared to LaDID. In addition, previous studies reported in literature outline the same weaknesses of the baseline methods and show almost identical results of future prediction of similar test cases, see for example Figs. 13, 22, 23 \& 24 in "Latent Neural ODEs with sparse Bayesian Multiple Shooting", Iakovlev et al., ICLR 2023.\
> These arguments support our view that the empirical results reflect real behaviour. Rather, we think the performance gap is due to the effectiveness of the decomposition we propose and study.
>
> Q7: Encoding of query time t\
> Thank you for this question. We represent each queried time step $t_q$ as a relative time embedding using a sin/cos encoding. Therefore, our dynamics model is continuous in time and, more importantly, can handle irregular time grid by default. Additionally, we are not forced to solve for intermediate time steps.

---

> > ### Comment · Reviewer_Fm8i · 2023-11-19
> > **reviewer feedback**
> >
> > I thank the authors for the rebuttal and clarification.
> >
> > q1: Upon checking the rebuttal response and other reviews, honestly I'm still confused about the definition of RS/RI. I don't agree with your reply that ODEs are not suited for complex or delayed systems because these delays can be treated as additional system states, or multiple steps can be concatenated as a single step to capture the delay. Taking that aside, since all the experiments in the paper are ODE/PDE-like dynamical systems, can you specify the physics meaning of RS/RI in one example and the difference between RS/RI to initial condition/system coefficients? I asked this question in the original review but not answered.
> >
> > q3: I checked the appendix this is not a convergence guarantee of what the method proposed to do (8oY3 also pointed out). There is neither intermediate experiment step to show how RS/RI decomposition works (e.g. how the embedding looks like wrt initial condition/system coefficients). It's difficult to justify the improvement of final results without intermediate steps.
> >
> > q5: why do you start drawing at 25% of the prediction steps? why don't start at the beginning to show how different methods diverge? given the information density of the current 13 drawing steps, there is plenty of room to accommodate the first 25%.
> >
> > q6: Regarding the method's effectiveness / massive improvement over prior methods, I'd like to hear the opinions from other reviewers.
> >
> > Given the rebuttal effort, I will increase my score to 5.

---

> > > ### Author Response · Authors · 2023-11-21
> > >
> > > Thank you for your continuing engagement with this work and your valuable feedback to improve the paper.
> > >
> > > Q1 \& Q3: We address these concerns in a global comment, as these points are of general interest, and hope that
> > > the further clarifications and new experiment address your question. The new experiment in Sec. K.8 of the Appendix supports the notion that the learned RS representation indeed
> > > carries information from which the true instance-specific parameters can be recovered, in line with the theory, and helping to explain the strong empirical performance when combined with a otherwise universal (non instance-specific) prediction module.
> > >  If you have any further questions or comments, we are happy to answer.
> > >
> > > Q5: Initially, we started to display prediction after 25\% since we put focus on long-horizon future prediction.  However, we updated all figures so that  they also display system states right from the beginning of the prediction. Please see Figs. 4 and K.1 – K.6 in the Appendix. The updated figures show that the baseline algorithms (except ODE-RNN) reconstruct reasonably the first few prediction time steps but diverge for long-term prediction.

---

### Official Review · Reviewer_8oY3 · 2023-10-28

**Soundness:** 2 fair
**Presentation:** 3 good
**Contribution:** 2 fair
**Rating:** 6
**Confidence:** 2

**Summary:**

The authors consider an encoder decoder model for predicting the trajectory of time series from high dimensional observations. A key difference of the model is that the authors do not impose a particular type of known structure on the dynamics such that the dynamics must follow this structure. The authors then introduce a probabilistic multiple shooting method such that the method can be trained. The authors illustrate the proposed method on a number of different datasets to validate its performance. The main question in the performance evaluation is how well the method works on unseen conditions, i.e. changing the parameters of the true underlying dynamical system.

**Strengths:**

The premise of the paper is very well founded. In general, most time series approaches with latent dynamics consider some model from the applied mathematics literature (e.g. SDEs, ODEs) and then parameterize it according to a neural network. However, the authors consider a more general framework where they split the dynamics according to realization-specific components and invariant components. This induces a new algebra that should somehow be given by the parameterization of the model. The empirical results are also very good and suggest that the model outperforms the baseline methods.

**Weaknesses:**

While the motivation of the paper is nice, there exist a number of limitations in the implementation and in the model itself. For one, there is no guarantee that the model will converge to something that has these separable parameter sets within the motivation of the work. The loss function seems rather standard in the sense that it is the standard loss when one parameterizes a latent variable time series model. It would have been nice to show under what assumptions does the model decompose into the disjoint parameter sets -- the invariant ones and the instance-specific ones.

While this is a small weakness, the paper seems to claim a bit too much (though I may have also misunderstood). For example, the authors mention they make no assumptions on the structure of the model in the sense of an ODE type of factorization, but still the fact that the first $K$ steps of the time series are used for conditioning does induce a particular structure. In some sense, the ODE interpretation is amenable since one can use the known properties of ODEs to probe the model.

A major goal is to generalize the output on a new domains where some invariant properties are maintained, but the experiments only demonstrated this in the case of changing parameters. However, maybe it would be helpful to see the behavior under shifts in the observation distribution. For example, could the distribution of the observation space change in a trivial way (e.g. different color map for the wave equation example) and could the distribution still be predicted?

There are also some properties that the authors impose in their loss (e.g. smoothness of the latent space) but that assumption seems to violate some of the motivating ideas behind the paper. Smoothness almost implies a differential equation in the latent space.

**Questions:**

Is there anything in the loss that guarantees that the factorization in terms of the RI and RS components? If not how can one impose this on the latent representation?

Assumptions on the latent space being Gaussian seem a bit restrictive, is there a reason for making this assumption? Specifically am wondering how this affects the motivation that very little structure should be imposed on the latent distribution. Will the smoothness and Gaussian terms will force the learned dynamics to resemble something that satisfies an ODE?

---

> ### Author Response · Authors · 2023-11-17
>
> Q1: Guarantee of decomposition through loss and graphical model\
> Thank you very much for your feedback. We have addressed your concerns in the overall response and hope you find this new insight satisfactory.
>
> Q2: Assumption on underlying system\
> Thank you for your feedback. By stating that we do not make assumptions on the underlying system, we wanted to stress that our approach is likewise suited for ODE and PDE-based system and in principle should also be applicable to longitudinal observations for which we do not know the governing equations since our framework is inspired by an amortized VAE (of course with various modifications). In this context, we also wanted to stress that LaDID does not require access to the underlying system variables (in contrast to neural operators) and can also deal with pure observations such as a video of the swinging single/double pendulum (instead of the angle/angular velocity) or the vorticity distribution in fluid mechanical test cases (instead of the velocity fields).
> As a result, LaDID requires access to a little number of consecutive time steps since a single observation might not provide sufficient information on the dynamics, for instance, based on a single video frame, we cannot tell in which direction the pendulum eventually is swinging.
>
> Q3: Generalizaton to color shifts/scaling\
> This is an interesting aspect which we originally wanted to outline by means of the second experiment in Section 6.2. Here, we shifted the cylinder location within the simulation which essentially represents a shift in the observational distribution.
>
> However, the idea of a color shift in the observations is certainly a more intuitive example. Therefore, we trained our approach on a swinging pendulum rendered in the RGB image space and modified the color space according to your suggestion. During training we applied standard image augmentations such as brightness, contrast, saturation and hue transformations. Results are shown in Sec. M2 of the Appendix and highlight that our approach can deal well with different color shifts.
>
> Q4: Properties of smoothness prior\
> In our work, the smoothness prior is used exclusively to ensure a smooth transition between subtrajectories when training under a multiple shooting augmentation. The smoothness prior does not impose any regularization on the form or shape of latent trajectories, but simply stitches together the last and first state of consecutive subtrajectories with no effect on the latent roll-out at inference. Therefore, we observed in our tests that the smoothness prior mainly stabilizes the training and improves the final evaluation performance slightly. That is, training without a smoothness eventually may also converge and yields valuable system predictions as shown in Tab. N.3 of the Appendix.
>
> Q5: Restrictions of Gaussian latent state prior\
> Thank you for your insightful question. The assumption of the latent space being Gaussian in our model is indeed a modeling choice that provides certain advantages, namely tractable inference and known effectiveness in various settings. The adoption of a Gaussian prior serves the purpose of establishing a gradient field within the latent space. Specifically, this prior acts to consolidate the estimated latent states $z_{t_q}$ and, in doing so, mitigates the emergence of empty regions devoid of meaningful semantics. This ensures that the latent trajectory roll-outs are well-defined under distribution shifts, noise, and other perturbations. While the Gaussian assumption introduces a level of smoothness, it does not necessarily force the learned dynamics to resemble a specific form, such as an ODE. This flexibility allows the model to capture complex and non-linear dynamics, making it a reasonable choice for many scenarios.

---

> > ### Comment · Reviewer_8oY3 · 2023-11-20
> >
> > Many thanks to the authors for their response and modifications in the manuscript. I appreciate all the experiments and the additional context. I have increased my score as a result.

---

> > > ### Author Response · Authors · 2023-11-21
> > >
> > > Thank you very much for your positive feedback and for revising your score. We are glad that the additional experiments and modifications have improved the manuscript. Your feedback has been invaluable, and we are grateful for your time and effort in the review process. If you have any further suggestions or questions, please feel free to let us know.

---

### Official Review · Reviewer_1fGG · 2023-10-30

**Soundness:** 3 good
**Presentation:** 4 excellent
**Contribution:** 2 fair
**Rating:** 6
**Confidence:** 3

**Summary:**

The authors propose a general framework for predicting dynamical trajectories based from data. The model architecture specificially aims at extracting realization-invariant and realization-specific information from different observations of underlying dynamical processes, iterated e.g. from different initial states or with different model parameters. To this end they employ a sophisticated encoder architecture, based on CNNs, temporal self-attention and a multi-shooting augmentation to obtain a condensed representation that can be used as a conditioning for a dynamics forecast. The learned latent dynamics model can be queried at irregular times, and the predicted dynamics decoded via a CNN-based decoder model and compared with the data. The model shows SOTA performance on 6 complicated and high-dimensional observations of dynamical processes, such as fluid flows. The author also investigate transfer learning between different parameter settings of the same underlying systems via few-shot learning.

**Strengths:**

The presentation, including the figures and writing, is high quality. I also appreciate the detailed appendix. While I have seen many of the components employed in similar settings, such as sequential variational autoencoders or Latent-ODEs, the proposed architecture looks like an interesting new combination of all of them. The idea of transfer learning and generalization for dynamics models is an important field of study. I like the approach of using an expressive encoder to inform the dynamics model about the specifics of the observed time series by encoding it in the initial state, and the fact that the obtained model is continuous in time.

**Weaknesses:**

General problem setting:

What is meant by invariant? To my understanding the term “invariant” in this paper mostly refers to the flow of the underlying system (i.e. the equations given in Appendix 1), while the realization-specific components are different initial conditions from which different observations start. This is also the situation primarily studied in the experimental section. The authors claim to show "generalization well beyond the data distributions seen during training".
However, the setting they mostly investigate is just the standard setting in dynamical systems reconstruction, where underlying dynamical processes/flow fields are inferred from multiple or even single observations. I don't think this constitutes generalization beyond the data distribution in the usual sense.

More interesting to my mind is the question of transfer learning e.g. between different dynamical regimes (as studied out in 6.2.) with different parameter settings of the underlying system, or even between different but in some way related systems. Relating this to a more formal language common in dynamical systems theory might be helpful, since for dynamical systems the concept of invariants, such as topological invariants, fixed points, periodic orbits, Lyapunov spectra, invariant measures etc. is well studied. While the abstract mentions the motivation to “ enforce certain scientifically-motivated invariances” these invariances are not really made explicit in the related work or in the analysis of the obtained results and remained somewhat elusive to me.
Further, while you mention that your approach “is in principle not specific to physical ODE-like models”, these are the only systems you actually study in the experiments, which naturally suggests are framing in these terms.

Transfer learning setting (6.2.)

Your framework is aimed at transfer learning from “potentially highly heterogeneous parameters and data”, which as mentioned seems the more novel application of your framework. Most of the empirical section however focuses on multiple instantionations of the same underlying system. The results from 6.2., where transfer learning between different model parameters is learned, however indicate that the transfer effect is relatively weka, with advantages over learning from scratch remaining within the error bars, and comparative performance requiring a significant portion of the new training set (if I understood it correctly corresponding to hundreds of tractories from the altered system).

Interventions/nonstationarities in parameters for dynamical systems can also be viewed through the lense of bifurcations. Increasing the Reynolds number can e.g. induce turbulence, as you rightly point out. From your description it is not altogether clear what changes in the dynamics your parameter changes in this section induce, and whether they lead to qualitative differences.
Since to my understanding this transfer is the primary motivation for introducing your framework, I found this section to be a little too short and and not in-depth enough.
I also wasn't 100% sure if you indeed jointly trained on different Reynolds numbers for the flow around a blunt body (Re = 100, 250, 500), which would be an interesting extension of your study in 6.2.

Comparisons and Experimental Evaluation

All datasets studied in the comparison methods are based on solutions to synthetic ODE systems, and include several PDEs. The claim that you provided evaluation on real-world datasets I find a little misleading, since none of the benchmarks are based on experimental/real world data, while of course being inspired by real systems.

Selecting comparisons is always a debatable point, but since all benchmark systems chosen are relatively simple/polynomial form, other classes of comparisons could be valuable: approaches by Brunton et al. based on SINDy and its variants are also frequently studies similar fluid flows. Neural Fourier Operators, which you also mention in the related work have been particularly successful for PDEs, so they might be a better choice here.

Algorithms like ODE-RNN are tailored towards irregularly sampled, but not necessarily spatially extended data. ODE-RNN in general has very poor performance everywhere in the paper (see all figures including ODE-RNN, e.g. Fig 4 or Fig. J6 b), where they seem to have learned nothing meaningful at all, which indicates that there either is a bug in the training, or the method is not well suited as a comparison.

Overall, since I think the general setting studied here is important and the presentation is good, if my concerns can be adressed I am also willing to adjust my final score.

**Questions:**

Model parameters:

How parameter intense is your approach? Would it struggle to infer meaningful dynamics from only single or few trajectories, compared to approaches with a stronger inductive bias, such as SINDy (Brunton et al., 2015), or experimental settings, where there are often only single or few observed trajectories. The models in Table H2 have between 1 and 10 million parameters, and are trained on observations with 400 initial states each used for training, which hints at a data and parameter intese regime given some of the relatively simple underlying dynamical processes. How does your model behave for sparser data? Figure 6 indicates that for comparable performance on the pendulum data, comparable performance requires hundreds of observed trajectories?

Related Work:

I understand that it's hard to provide an exhaustive overview over related methods, and I appreciate the extended related work. I however want to point out that there are many RNN based approaches specifically designed for inferring complex dynamics and leveraging inferred models for long-term time series forecasting (Reservoir Computing, see Pathak et al. 2018), LSTMs (Vlachas et al., 2018) or RNNs (see e.g. Hess et al., ICML 2023), and could be mentioned here as well.

---

> ### Author Response · Authors · 2023-11-17
>
> Q1: Meaning of invariances and generalization\
> Thank you for your thoughtful questions and we appreciate the opportunity to provide further clarification. Please see the overall response, which we think addresses these points. Regarding generalization: what we mean is that different instances can lead to very different data distributions and therefore cannot be handled by straightforward supervised learning, for instance.
>
> Q2: Boundaries of transfer learning using concepts of invariant measures\
> It would indeed be interesting to study the behaviour of LaDID's data driven approach on data from systems with known, specific, complex invariances. This goes beyond the scope of the present paper. We emphasize also that the invariances that we exploit are very general (and common to most areas of science), rather than rooted in mathematical properties of specific dynamical models.
>
> Q3: Performance of transfer learning / Joint experiment with different Reynolds numbers\
> Thank you very much for your thoughtful comment. However, we respectfully disagree with your conclusion regarding the perceived weakness of the transfer effect. In Section 6.2, we deliberately focus on two distinct types of interventions: (i) those on the parameters of the underlying system leading to a different trajectory and (ii) interventions on the perception of the dynamical system (noise or camera shifts). Our results in Section 6.2 strongly indicate that transfer learning, in both cases, exhibits good performance.
>
> Qualitative interventions on the dynamical system:\
> Fig. 6a illustrates that fine-tuning a pre-trained LaDID with just 8\% of the data size (32 new trajectories) achieves a performance level that matches the best state-of-the-art competitor, MSVI, which was trained on a dataset 12.5 times larger for the new system (see Fig 3).\
> To address your requested transfer learning experiment at different Re-numbers, we now include a series of transfer learning experiments where we jointly train on a dataset with Re-numbers of $Re=[100, 250, 500]$ and present predictions for a set of Re-numbers of $Re=[100, 175, 250, 300, 400, 500]$. Please see Sec. P in the Appendix for detailed results.
>
> Interventions upon the measurement process:\
> Fig. 6b presents the effects of interventions upon the camera position when training on a dataset with 160 trajectories and Re-numbers of $Re=\{100, 250, 500\}$, however the underlying flow are not affected by the interventions.  Furthermore, we added Sec. M to the Appendix to illustrate the effect of perturbations commonly occurring during measurements, e.g. noise or systematic perception errors.
>
> In our opinion, the results in Sec. 6.2 of the main text and Sec. M and P demonstrate a robust transfer effect. We hope that these additional details clarify the effectiveness of our framework in handling transfer learning tasks.
>
> Q4: Selection of baselines and experiments\
> We appreciate your perspective on the choice of datasets and comparison algorithms, and we are open to exploring new suggestions. It is correct that our datasets are simulations rather than experimental measurements. However, we carefully selected simulations for real-world systems across various disciplines, which play crucial roles in typical engineering tasks. These systems are both nontrivial and allow us to properly explore model behavior.
>
> Regarding the choice of baseline algorithms, we established two key requirements: (i) the algorithm must involve some form of latent representation, and (ii) it must be continuous in time. While requirement (i) is restrictive, we believe it is necessary to focus on approaches that include dimension reduction. For that reason, we intentionally excluded interesting approaches like SINDy, FNO, or GPODE, as they operate directly in the observational space. Nevertheless, we consider our chosen baseline algorithms as strong competitors. With the exception of ODE-RNN, they consistently demonstrate reasonable predictions in most test cases. We agree that the original application of ODE-RNN was not for spatio-temporal systems. Instead, we adapted it by using our encoder module to generate latent input embeddings and utilized the MSVI decoder for a meaningful comparison. While ODE-RNN may not be the ideal benchmark, we included it for its foundational theoretical and practical relevance to the other benchmarks.
>
> Q5: Performance with reduced number of training trajectories\
> For a more in-depth analysis, we have updated Section K.7 in the Appendix, introducing a series of experiments involving different numbers of training trajectories. The outcomes depicted in Fig. K7 demonstrate that utilizing only 16\% of the training trajectories (64 trajectories) attains a performance level comparable to MSVI trained on the complete dataset.
>
> Q6: Extension of related work\
> We have added a dedicated section to the extended related work, where we discuss these relevant RNN-based approaches.

---

> ### Comment · Reviewer_1fGG · 2023-11-20
>
> I thank the authors for their detailed rebuttals and additions to the experimental evaluation and related work.
>
> Several referees agreed with the somewhat vague conception of the separation of RI and RS components. While the comments have aimed at clarifying this, I agree with Reviewer Fm8i that this should be elaborated on more clearly and give a concrete example, especially given the close similarity of the experimental section to the distinction "model parameters" as RI and "initial states" as RI.
>
> As per Fm8i's comment, I also found the strong performance difference between methods a little hard to understand, since the selected benchmark systems seem fairly standard, and the proposed approach is not specifically tailored to excel at this kind of PDE learning.
>
> This also connects my question w.r.t. the transfer learning effect between dynamical regimes, which is not so much about comparison to other approaches, as explored in the new figure in K7, but more about how well out-of-domain generalization works without retraining on significant portions of the novel train set, or even with zero-shot predictions.
> I appreciate the new experiments in section P, which go in this direction. I completely understand that new results hard to achieve during the revision under time pressure, but the way they are currently presented I find them a little difficult to interpret. Parts of Fig. P1b) are basically impossible to read, and a) I find a little challenging to interpret. Given as I mentioned I think this might be one of the more important applications of your framework, I would appreciate a clearer treatment and analysis in case the paper is accepted.
>
> In light of the improvements and detailed feedback I am also happy to adjust my score and tend towards accepting.

---

> > ### Author Response · Authors · 2023-11-21
> >
> > Thank you for your thorough evaluation and the recognition of our efforts during revision. We appreciate your feedback concerning better understanding RS/RI separation. We have addressed this point in our second global comment with a new experiment (see Section K.8 in revised paper). We think that the new results support the notion that the learned RS representation indeed
> > carries information from which the true instance-specific parameters can be recovered, in line with the theory.
> >
> > Regarding the strong performance difference between methods and the question of out-of-domain generalization in transfer learning,
> > thank you for drawing our attention to shortcomings in the presentation. In the revised manuscript, we have improved the readability of Fig. P1a and P1b. We understand the significance of these aspects, especially in the context of the broader applications of our framework, and we are open to every comment and suggestion to ensure a clear and accessible presentation.
> >
> > We appreciate your flexibility in adjusting your score and your willingness to consider acceptance in light of the improvements. Thank you very much.

---

### Official Review · Reviewer_apHo · 2023-10-31

**Soundness:** 3 good
**Presentation:** 2 fair
**Contribution:** 3 good
**Rating:** 6
**Confidence:** 3

**Summary:**

The manuscript introduces a novel deep trajectory inference model, referred to as LaDID. This model designed to predict dynamical trajectories by disentangling invariant and variant dynamical factors. LaDID employs a convolutional autoencoder unit for dimension reduction and a transformer unit for learning temporal structure, eliminating the need for ODE formulations and enabling efficient long trajectory prediction without added computational costs.

In the study, the authors evaluated LaDID's performance using six simulated datasets. They conducted a comprehensive comparison of LaDID with four ODE-based models, namely ODE-RNN, ODE2VAE, NDP, and MSVI. The results demonstrated LaDID's superiority in all experiments, including challenging few-shot learning tasks.

I appreciate the problem tackled by the author and find the concept of invariant-latent representation learning in dynamical systems intriguing. The paper appears to have a solid metric in place. Nevertheless, there are certain crucial aspects that need to be improved. I am open to potentially revising my assessment following the author's rebuttal.

**Strengths:**

- The manuscript tackles an intriguing and challenging problem related to trajectory inference in dynamical systems. The proposed approach involves effectively decomposing the latent factors into variant and invariant components, which is fairly innovative.

- The authors conducted a comprehensive performance evaluation across six complex dynamical systems.

- The proposed model consistently outperforms its related counterparts across a wide range of tasks.

- The proposed model showcases capabilities in handling long trajectories, even when provided with limited training samples. This adaptability to data constraints is a significant strength, as it opens up opportunities for more efficient and practical application in dynamical system modeling.

**Weaknesses:**

**Major comment:**

My prominent concern in the paper pertains to the insufficient explanation for why their model outperforms others. The main contribution of the paper lacks clear description and justification. As pointed out by the authors, both LaDID and MSVI (Iakovlev et al., 2023) share several similarities, such as using CNN encoders/decoders and the same transformer. The primary distinction between these models lies in replacing neural ODE with MLPs to learn $f_{\phi_{dyn}}$ and introducing a smoothness loss to discourage abrupt transitions between latent subpatches. It remains unclear how these modifications enhance the process of learning latent dynamics in comparison to ODE-based methods.

**Minor Points:**

- Figure 1 could benefit from a more comprehensive overview of their framework. Clarifying how each process in the top row corresponds to each step in the bottom row would enhance understanding.
- The paper's writing can be improved, and some content from the Appendix may be integrated into the main text.
- The descriptions of the experiments are somewhat confusing and could benefit from increased clarity and explanation.

**Questions:**

- If I understand correctly, the paper addresses only the initial state as the specific (variant) trajectory factor, right? What about other sources of variability, like measurement noise?

- In section 4.1, Model, the main text says *"To obtain a latent trajectory,  ... we choose as a set of different sine and cosine waves with different wave length."* Could you explain it?

- Could you please explain the choice of using MSE/RMSE for evaluation? It seems that capturing the overall structure of dynamics (direction, shape, etc.) might be more crucial than the absolute amount of misprediction.

- Can LaDID effectively handle time-variant dynamical systems? If yes, how?

- In the graphical model (Figure A1), $\psi^r$ is out of the box "$N$". Then, how is $\psi^r$ a function of "$n$" in Eq. 3 and Eq. 9?

- The text mentions *" ... mean-aggregation, which can be changed based on the task at hand."* Could you elaborate on the role and task-dependent nature of mean-aggregation?

- In Eq. 9, within the representation prior term, none of the variables appear to be a function of "$n$." Can you clarify this discrepancy?

- What does "$D$" represent in Figure A1?

- The authors stated that *"our models are specifically designed to exploit certain invariances that are important in classical scientific models."* Could you provide more detail on these "certain invariances"?

- Beyond classical scientific examples, can LaDID effectively learn developmental trajectories involving bifurcations and other complex phenomena?

---

> ### Author Response · Authors · 2023-11-17
>
> Q1: Unclear how modifications enhance learning of latent dynamical systems\
> Please find a fuller account of how our approach enhances the learning of latent dynamics in the overall response above. We clarify there why our approach is different from e.g. MSVI, and explain the invariances in more detail. We emphasize that they are general properties that we think hold for a variety of systems rather than specific invariances derived from equations for a particular system.
>
> Q2: Fig 1 - more comprehensive overview\
> Thank you for the suggestion; we have modified figure 1 to improve the visualization of our network.
>
> Q3: Improved writing, add parts from App. to MS\
> We are highly motivated to improve the clarity of our work and would appreciate any direct feedback on unclear sections. However, we are constrained by the strict page limit, which limits the amount of additional content we can include.
>
> Q4: Improved Exp. Description\
> Due to the page limit, we have added the Sec. J in the Appendix to provide a more comprehensive description of the experiments and hope that the updated Appendix addresses your concerns.
>
> Q5: Measurement noise\
> To demonstrate the robustness to noise, we provide evaluation metrics for various noise levels added to the mean of our experiments, e.g. Gaussian noise ($\sigma = [0.1, 0.2, 0.3, 0.4]$) and salt-and-pepper noise ($\rho = [10\\%, 35\\%, 50\\%, 70\\%]$). The results are presented in Sec. M1 and M2 of the Appendix. We note also that within our loss-based framework, measurement and process noise are, in a way, handled under a unified overall objective.
>
> Q6: Encoding of time queries\
> We stack the sin/cos of the first 16 integer frequencies contained in the observed time interval similar to a Fourier series yielding a 32-dimensional relative time encoding.
>
> Q7: Why MSE for evaluation?\
> We agree that time-averaged RMSE values do not capture all relevant information and can easily yield misleading interpretation. Therefore, we consider three metrics including the time-averaged nMSE, the RMSE over time and the visualization of the pixelwise $L2$-error (see Figs J1-6) allowing us to precisely study the accuracy of the predicted dynamics.
>
> Q8: Can LaDID effectively handle time-variant dynamical systems?\
> Since we have not conducted experiments for such systems, we are hesitant to make claims about the effectiveness of LaDID in this context. However, we are optimistic that LaDID could be effective for some such systems, provided suitable higher-level regularities exist. Here, a key point is that LaDID does not pre-suppose any particular kind of model but rather the high-level properties detailed above, which can hold for some time-varying systems. The observed generalization performance in Section 6.2 suggests that LaDID may have the potential to handle time-varying systems. Further investigation will be required to address this question in future work.
>
> Q9: Notation in graphical model vs. eq. 3 and 9\
> Thank you for bringing this to our attention. It is important to distinguish between the conceptual representation in Fig. A1 and the implementation of the LaDID model. In Fig. A1, we illustrate the overarching concept, indicating that for each trajectory, we generate a unique $\psi^r$ and subsequently roll out the latent trajectory. The plate notation $N$ represents the length of the requested trajectory points. In eq. 3 and 9, the variable $N$ denotes the number of subtrajectories, with the understanding that the concept outlined above holds true for each subtrajectory. We recognize the potential confusion and updated the notation in Figure A1 and its caption for clarity.
>
> Q10: Mean-aggregation and possible alternatives\
> In the proposed implementation we choose the mean for sake of simplicity, but many different aggregations appear possible. One counterpart which we want to investigate in future work is the replacement of the mean by a singular value decomposition. Since the derived embeddings of the $k$ input samples are low-dimensional, a SVD may be cost efficient in extracting coherent data structures across all embeddings. Another promising replacement may be the introduction of a RNN where we want to use its final hidden state as general initial condition representation.
>
> Q11: Dependence of prior on subtrajectory\
> All priors are computed per subtrajectory and thus a function of $n$ which we assumed to be clear from the context, but for clarity, we have added the dependence to the equations.
>
> Q12: Graphical model - plate notation D\
> In Fig. A1, D refers to the total number of the training samples. We amended the figure caption to clarify this aspect.
>
> Q13: Application to bifucations and other phenomena\
> We appreciate your curiosity in this regard but at the stage, we lack evidence to make statements about LaDID's effectiveness in learning such dynamics. We plan to conduct thorough investigations into the performance of LaDID in such scenarios in future.

---

> > ### Comment · Reviewer_apHo · 2023-11-21
> > **Response to the rebuttal**
> >
> > I appreciate the author's efforts in presenting a compelling rebuttal alongside additional experiments. The supplementary experiments, particularly section M, offer intriguing insights into the model's capabilities.
> >
> > However, before finalizing my decision, I feel it's crucial to revisit my primary concern regarding the outperformance of the proposed model. Despite the comprehensive responses addressing various aspects in the overall response, I still seek clarification on how LaDID significantly enhances learning ODE/PDE-based dynamics compared to the previous ODE-based approach. I fully acknowledge the potential for isolating invariant structures/mechanisms and the emphasis on the RS-RI decomposition to enhance inference. However, it remains hard to attribute all improvements solely to this factorization without demonstration of the level of decomposition achieved.
> >
> > Echoing the concerns of other reviewers, I believe the authors should provide quantitative evidence to support the model's superior performance by quantifying the extent of disentanglement between RI and RS. For a given experiment, explicitly defining RS and RI parameters and demonstrating how effectively the model infers these parameters would significantly strengthen the argument for the model's effectiveness.

---

> > > ### Author Response · Authors · 2023-11-21
> > >
> > > Thank you for your continued engagement with our work and for recognizing the efforts in our rebuttal and additional experiments! We appreciate your attention to detail and understand your question concerning better understanding the RS-RI decomposition.
> > > We have added material to the global comment to clarify and address this point and an additional experiment to the revised paper (see global comment for details).
> > >
> > > We are committed to addressing this question comprehensively, so please let us know if any further questions might arise. We hope that this additional analysis will address your questions and contribute to a deeper understanding of LaDID's behaviour in learning dynamics.

---

### Author Response · Authors · 2023-11-17

We appreciate the reviewers' careful reading and insightful comments. The paper has been thoroughly revised in response, resulting in significant improvement.

We start with some comments on the invariances we seek to exploit. Our general idea is to exploit the notion that for a given class of system, a kind of universal law can explain different instances (of the same type of system), even when the {\it data} for the instances appears very different. This general motivating notion in science underlies our formal decomposition into realization-specific (RS) and realization-invariant (RI) information and underpins both our theoretical results and the proposed architecture. As a result, our framework can apply to both deterministic and stochastic models, as well as temporal and spatio-temporal systems.

Typically, in differential equations, the interplay of functional invariances and boundary/initial conditions determines the unique solution. In contrast our scheme is more general and rooted in a loss-based view. The exploited invariances  are very general rather than specific to any particular kind of dynamical model and our architecture reuses existing tools and modules (transformers etc.). The strong empirical performance -- even relative to methods using similar modules -- supports the power of exploiting these invariances. While  at first glance the approach may appear similar to various competitors (using related neural modules), we think it is important to recognize that the underlying idea (and associated graphical model, see Fig A1) is novel, fundamentally different and highly effective.

A key aspect is the strict separation of RS and RI components, achieved through careful design in our learning framework. This separation has significant implications for the latent dynamical model and its overall performance.
Our method is data-driven and does not require identification of the expected system equations. This stands in contrast to much of the classical dynamical systems literature, where the focus is often on invariant/convergent behaviors derived from specific models.

Our RS/RI decomposition is more closely related to the notion of invariant mechanisms, as studied by Peters et al (see our graphical model in Fig A1). Specifically, the absence of a directed connection between the parameters $\Phi$ of the RI dynamical model and the RS trajectory representation $\psi^r$ implies two  aspects: (i) (marginal) independence between them, and (ii) the RS trajectory representation $\psi^r$ must contain all instance-specific information to roll-out the entire latent trajectory.
The universal model parameters $\Phi$ in that sense have to contain the entire underlying 'law' of the system, while the representation $\psi^r$ is the only place RS information is allowed to be stored. Note also that  predicted latent states $z_{t_{q_i}}$ and $z_{t_{q_j}}$ are conditionally independent (c.i.) given the parameters $\Phi$ of the RI dynamical model and the RS trajectory representation $\psi^r$, i.e. $P(z_{t_{q_i}}|\Phi, \psi^r) \perp\perp P(z_{t_{q_j}}|\Phi, \psi^r), ~i\neq j$.
Naturally, the marginal dependence of latent states is inherent, as the c.i. relation arises from the information encapsulated in $\Phi$ and $\psi^r$. Consequently, there is no requirement for a Markov assumption at the latent state level ($z_{t_{q-1}}$ is not required to predict $z_{t_{q}}$).

The theoretical results in the paper show that under certain assumptions (i.e. the sufficient encoding/mapping assumptions), such a decomposition can be learned. Importantly, we do not explicitly identify the elements of the overall function, relying instead on black-box neural networks for approximation. Acknowledging the non-trivial nature of finding the global optimum, we employ stochastic gradient descent, a widely used approach known for converging to local minima, consistent with established practice in neural network applications. We recognize the importance of expressiveness in these networks, and the experiments demonstrate significant improvements over multiple competitive baselines across a broad range of real-world-inspired examples.

Our model emphasizes generalization by requiring a single realization-invariant (RI) model
to predict {\it all} trajectories. We do not claim that LaDID can predict the progression of any arbitrary system, rather we assume that input data blocks are indeed different instances of the same kind of system. That said, our results include examples showcasing strong generalization capabilities to systems sharing to some extent similarities with the training distribution.


We hope these clarifications resolves any ambiguity regarding the use of invariances, the novelty of our approach and the entailed differences to other approaches. An anonymous code repository is available under:
https://github.com/kl844477/LaDID

---

### Author Response · Authors · 2023-11-21
**Global response 2**

We thank all reviewers for their continuing engagement and input to improve the manuscript. We  appreciate the insightful comments and greatly value the opportunity to reflect on and clarify the key ideas.
We understand that the aspect of RS/RI decomposition is both  important but also in some ways subtle hence we provide below some additional comments on this point  in response to feedback received.

* In our theory, the learned RS term $\psi^r$ is assumed to admit a correction to the true RS term and this turns out to imply learnability (see paper). However, we do not  obtain the correction explicitly
and therefore do not claim in general to be able to estimate the true RS term (which for some physical examples might be an initial condition/system coefficient). Rather, the claim is that $\psi^r$ is sufficient to (in principle) recover this information and (loosely speaking) this is why prediction is possible using an otherwise universal model. Adding constraints to e.g. guarantee identifiability of the true initial conditions would limit the scope or constrain us to specific settings with strong prior knowledge. In this context, it is relevant to note that while $\psi^r$ may be interpreted as initial conditions in cases of ODE-based systems, the concept is more general and essentially seeks to capture any instance-specific information, which could be different in different examples (PDEs, SDEs, real-world systems that do not map perfectly to standard formalisms etc.). For the general case, our framework seeks to capture instance-specific information in
a  representation
$\psi^r$ to then allow evolution over time using
the RI model.
In a way, this allows various
different kinds of systems with different
instance-specific boundary conditions and system parameters to be dealt with in a unified, risk-based framework.

* As noted above, we do not guarantee recovery of the true initial conditions/system coefficients. Therefore, to better understand model intermediates, as suggested in the latest comments received, we carried out an additional experiment tracking the learned terms $\psi^r$ over a set of pendulum configurations for which we know the true coefficients $\pi_r$. In case of a successful RS/RI separation, it ought be possible to recover the latter from the former. To test this we trained a very simple and light NN (2 layer MLP) on inputs $\psi^r$ and indeed we find that the $\pi_r$'s can be recovered (the simple NN is in effect an empirical estimate of the *correction function* that appears in the theory). We chose the single pendulum dataset as the initial conditions for this kinematic benchmark are easy to understand, i.e. $\pi_r = ($angle $\theta,$ angular velocity $\dot{\theta})$.
For these additional experiments, we use a trained LaDID model and compute the RS trajectory representations $\psi^r$ for every initial condition of the training dataset (see Appendix K.8 in our revised paper for details). We present the results as scatter plots in Fig. K.8; the ability to indeed recover the true system parameters
supports the RS/RI theory and notion of a correction function.



We think these additional results provide further insight into *why* LaDID works:
in short, LaDID exploits the RS/RI structure to lead to a tractable learning problem via an invariant/universal prediction module combined with estimated RS information. As we show in the paper, this perspective leads to clear gains in challenging problems that remain nontrivial for SOTA neural-dynamical models.

---

### Meta-Review · Area_Chair_8gEm · 2023-12-09

**Metareview:**

This paper presents LaDID, a novel deep learning model for predicting dynamical trajectories. LaDID's key innovation is its ability to disentangle invariant (universal) and variant (system-specific) dynamical factors, enabling efficient long-term prediction without dependence on ODE formulations. The authors demonstrate LaDID's superiority over existing ODE-based models across various complex datasets, including challenging few-shot learning tasks.

While certain aspects require further improvement, the reviewers generally think the paper presents a solid foundation and intriguing concept. LaDID innovatively decomposes latent factors into variant and invariant components, achieving superior performance across diverse tasks in six complex dynamical systems. LaDID showcases its strength in handling long trajectories even with limited training data, opening up opportunities for efficient and practical application in dynamical system modeling. The paper's high-quality presentation further highlights the promising potential of LaDID in the field of transfer learning and generalization for dynamics models.

**Justification For Why Not Higher Score:**

While LaDID presents an interesting approach to learning dynamical systems, several critical concerns need to be addressed before the final publication. These include clarifying the model's explanation, strengthening the evidence for generalization and transfer learning, clarifying definitions, conducting a more thorough literature review, and providing concrete evidence for the claimed novelty and performance gains.

**Justification For Why Not Lower Score:**

While certain aspects require further improvement, the reviewers generally think the paper presents a solid foundation and intriguing concept. LaDID innovatively decomposes latent factors into variant and invariant components, achieving superior performance across diverse tasks in six complex dynamical systems. LaDID showcases its strength in handling long trajectories even with limited training data, opening up opportunities for efficient and practical application in dynamical system modeling. The paper's high-quality presentation further highlights the promising potential of LaDID in the field of transfer learning and generalization for dynamics models.

---

### Decision · Program_Chairs · 2024-01-16

Accept (poster)